# Freya PAGE: First Optimal Time Complexity for Large-Scale Nonconvex Finite-Sum Optimization with Heterogeneous Asynchronous Computations

**Alexander Tyurin**
KAUST,[*] AIRI,[†] Skoltech[‡]

**Kaja Gruntkowska**
KAUST[*]

**Peter Richtárik**
KAUST[*]

## Abstract

In practical distributed systems, workers are typically not homogeneous, and due to differences in hardware configurations and network conditions, can have highly varying processing times. We consider smooth nonconvex finite-sum (empirical risk minimization) problems in this setup and introduce a new parallel method, Freya PAGE, designed to handle arbitrarily heterogeneous and asynchronous computations. By being robust to "stragglers" and adaptively ignoring slow computations, Freya PAGE offers significantly improved *time complexity* guarantees compared to all previous methods, including Asynchronous SGD, Rennala SGD, SPIDER, and PAGE, while requiring weaker assumptions. The algorithm relies on novel generic stochastic gradient collection strategies with theoretical guarantees that can be of interest on their own, and may be used in the design of future optimization methods. Furthermore, we establish a lower bound for smooth nonconvex finite-sum problems in the asynchronous setup, providing a fundamental time complexity limit. This lower bound is tight and demonstrates the optimality of Freya PAGE in the large-scale regime, i.e., when $\sqrt{m} \geq n$, where $n$ is # of workers, and $m$ is # of data samples.

## 1 Introduction

In real-world distributed systems used for large-scale machine learning tasks, it is common to encounter device heterogeneity and variations in processing times among different computational units. These can stem from GPU computation delays, disparities in hardware configurations, network conditions, and other factors, resulting in different computational capabilities and speeds across devices [Chen et al., 2016, Tyurin and Richtárik, 2023]. As a result, some clients may execute computations faster, while others experience delays or even fail to participate in the training altogether.

Due to the above reasons, we aim to address the challenges posed by device heterogeneity in the context of solving finite-sum nonconvex optimization problems of the form

$$\min_{x \in \mathbb{R}^d} \left\{ f(x) := \tfrac{1}{m} \sum_{i=1}^{m} f_i(x) \right\}, \tag{1}$$

where $f_i : \mathbb{R}^d \to \mathbb{R}$ can be viewed as the loss of a machine learning model $x$ on the $i^{\text{th}}$ example in a training dataset with $m$ samples. Our goal is to find an $\varepsilon$-stationary point, i.e., a (possibly random) point $\hat{x}$ such that $\mathbb{E}[\|\nabla f(\hat{x})\|^2] \leq \varepsilon$. We focus on the homogeneous distributed setup:

- there are $n$ *workers/clients/devices* able to work in parallel,

---

[*]King Abdullah University of Science and Technology, Thuwal, Saudi Arabia

[†]AIRI, Moscow, Russia

[‡]Skolkovo Institute of Science and Technology, Moscow, Russia

38th Conference on Neural Information Processing Systems (NeurIPS 2024).

- each worker has access to stochastic gradients $\nabla f_j$, $j \in [m]$,
- worker $i$ calculates $\nabla f_j(\cdot)$ in less or equal to $\tau_i \in [0, \infty]$ seconds for all $i \in [n], j \in [m]$.

Without loss of generality, we assume that $\tau_1 \leq \ldots \leq \tau_n$. One can think of $\tau_i \in [0, \infty]$ as an upper bound on the computation time rather than a fixed deterministic time. Looking ahead, iteration complexity can be established even if $\tau_i = \infty$ for all $i \in [n]$ (Theorem 4). We also provide results where the bounds $\{\tau_i^k\}$ are dynamic and change with every iteration $k$ (Section 4.4). For simplicity of presentation, however, we assume that $\tau_i^k = \tau_i$ for $i \in [n], k \geq 0$, unless explicitly stated otherwise.

## 1.1 Assumptions

We adopt two weak assumptions, which are standard for the problem (1) [Fang et al., 2018].

**Assumption 1.** *The function $f$ is $L_-$-smooth and lower-bounded by $f^* \in \mathbb{R}$.*

**Assumption 2.** $\exists L_+ \geq 0$ *such that* $\frac{1}{m} \sum_{i=1}^{m} \|\nabla f_i(x) - \nabla f_i(y)\|^2 \leq L_+^2 \|x - y\|^2 \ \forall x, y \in \mathbb{R}^d$.

We also consider Assumption 3. Note that this assumption does not restrict the class of considered functions $\{f_i\}$. Indeed, if Assumption 2 holds with $L_+$, then Assumption 3 holds with some $L_\pm \leq L_+$. If one only wants to rely on Assumptions 1 and 2, it is sufficient to take $L_\pm = L_+$. However, Assumption 3 enables us to derive sharper rates, since $L_\pm$ can be small or even 0, even if $L_-$ and $L_+$ are large [Szlendak et al., 2021, Tyurin et al., 2023, Kovalev et al., 2022].

**Assumption 3** (Hessian variance [Szlendak et al., 2021]). *There exists $L_\pm \geq 0$ such that*

$$\frac{1}{m} \sum_{i=1}^{m} \|\nabla f_i(x) - \nabla f_i(y) - (\nabla f(x) - \nabla f(y))\|^2 \leq L_\pm^2 \|x - y\|^2 \qquad \forall x, y \in \mathbb{R}^d.$$

## 1.2 Gradient oracle complexities

Iterative algorithms are traditionally evaluated based on their *gradient complexity*. Let us present a brief overview of existing theory. The classical result of Gradient Descent (GD) says that in the smooth nonconvex regime, the number of oracle calls needed to solve problem (1) is $\mathcal{O}(m\varepsilon^{-1})$ because GD converges in $\mathcal{O}(\varepsilon^{-1})$ iterations, and calculates the full gradient $\nabla f = 1/m \sum_{i=1}^{m} \nabla f_i$ in each iteration. This was improved to $\mathcal{O}(m + m^{2/3}\varepsilon^{-1})$ by several *variance-reduced* methods, including SVRG and SCSG [Allen-Zhu and Hazan, 2016, Reddi et al., 2016, Lei et al., 2017, Horváth and Richtárik, 2019]. Since then, various other algorithms, such as SNVRG, SARAH, SPIDER, SpiderBoost, PAGE and their variants, have been developed [Fang et al., 2018, Wang et al., 2019, Nguyen et al., 2017, Li et al., 2021, Zhou et al., 2020, Horváth et al., 2022]. These methods achieve a gradient complexity of $\mathcal{O}(m + \sqrt{m}\varepsilon^{-1})$, matching the lower bounds [Fang et al., 2018, Li et al., 2021].

That said, in practical scenarios, what often truly matters is the *time complexity* rather than the *gradient complexity* [Tyurin and Richtárik, 2023]. Although the latter metric serves as a natural benchmark for sequential methods, it seems ill-suited in the context of parallel methods.

## 1.3 Some previous time complexities

Let us consider some examples to provide intuition about time complexities for problem (1).

GD **with** 1 **worker** (Hero GD). In principle, each worker can solve the problem on their own. Hence, one approach would be to select the fastest client (assuming it is known) and delegate the task to them exclusively. A well-known result says that for $L_-$-smooth objective function $f$ (Assumption 1), GD converges in $\delta^0 L_- \varepsilon^{-1}$ iterations, where $\delta^0 := f(x^0) - f^*$, and $x^0$ is the starting point. Since at each iteration the method computes $m$ gradients $\nabla f_i(\cdot)$, $i \in [m]$, the time required to find an $\varepsilon$-stationary point is $\delta^0 L_- \varepsilon^{-1} \times m\tau_1$ seconds.

GD **with** $n$ **workers and equal data allocation** (Soviet GD). The above strategy leaves the remaining $n - 1$ workers idle, and thus potentially useful computing resources are wasted. A common approach is to instead divide the data into $n$ equal parts and assign one such part to each worker, so that each has to compute $m/n$ gradients (assuming for simplicity that $m$ is divisible by $n$). Since at each iteration the strategy needs to wait for the slowest worker, the total time is $\delta^0 L_- \varepsilon^{-1} \times m\tau_n/n$. Depending

Table 1: Comparison of the *worst-case time complexity* guarantees of methods that work with asynchronous computations in the setup from Section 1 (up to smoothness constants). We assume that $\tau_i \in [0, \infty]$ is the bound on the times required to calculate one stochastic gradient $\nabla f_j$ by worker $i$, $\tau_1 \leq \ldots \leq \tau_n$, and $m \geq n \log n$. Abbr: $\delta^0 := f(x^0) - f^*$, $m =$ # of data samples, $n =$ # of workers, $\varepsilon =$ error tolerance.

| Method | Worst-Case Time Complexity | Comment |
|---|---|---|
| Hero GD   (Soviet GD) | $\tau_1 m \frac{\delta^0}{\varepsilon} \quad \left( \tau_n \frac{m}{n} \frac{\delta^0}{\varepsilon} \right)$ | Suboptimal |
| Hero PAGE   (Soviet PAGE) [Li et al., 2021] | $\tau_1 m + \tau_1 \frac{\delta^0}{\varepsilon} \sqrt{m} \quad \left( \tau_n \frac{m}{n} + \tau_n \frac{\delta^0}{\varepsilon} \frac{\sqrt{m}}{n} \right)$ | Suboptimal |
| SYNTHESIS [Liu et al., 2022] | — | Limitations: bounded gradient assumption, calculates the full gradients[a], suboptimal.[b] |
| Asynchronous SGD [Koloskova et al., 2022] [Mishchenko et al., 2022] | $\frac{\delta^0}{\varepsilon} \left( \left( \sum\limits_{i=1}^{n} \frac{1}{\tau_i} \right)^{-1} \left( \frac{\sigma^2}{\varepsilon} + n \right) \right)$ | Limitations: $\sigma^2$–bounded variance assumption, suboptimal when $\varepsilon$ is small. |
| Rennala SGD [Tyurin and Richtárik, 2023] | $\frac{\delta^0}{\varepsilon} \min\limits_{j \in [n]} \left( \left( \sum\limits_{i=1}^{j} \frac{1}{\tau_i} \right)^{-1} \left( \frac{\sigma^2}{\varepsilon} + j \right) \right)$ | Limitations: $\sigma^2$–bounded variance assumption, suboptimal when $\varepsilon$ is small. |
| Freya PAGE (Theorems 7 and 8) | $\min\limits_{j \in [n]} \left( \left( \sum\limits_{i=1}^{j} \frac{1}{\tau_i} \right)^{-1} (m + j) \right) + \frac{\delta^0}{\varepsilon} \min\limits_{j \in [n]} \left( \left( \sum\limits_{i=1}^{j} \frac{1}{\tau_i} \right)^{-1} (\sqrt{m} + j) \right)$[c] | Optimal in the large-scale regime, i.e., $\sqrt{m} \geq n$ (see Section 5) |
| Lower bound (Theorem 10) | $\min\limits_{j \in [n]} \left( \left( \sum\limits_{i=1}^{j} \frac{1}{\tau_i} \right)^{-1} (m + j) \right) + \frac{\delta^0}{\sqrt{m} \varepsilon} \min\limits_{j \in [n]} \left( \left( \sum\limits_{i=1}^{j} \frac{1}{\tau_i} \right)^{-1} (m + j) \right)$ | — |

Freya PAGE has *universally* better guarantees than all previous methods: the dependence on $\varepsilon$ is $\mathcal{O}\left( 1/\varepsilon \right)$ (unlike Rennala SGD and Asynchronous SGD), the dependence on $\{\tau_i\}$ is harmonic-like and robust to slow workers (robust to $\tau_n \to \infty$) (unlike Soviet PAGE and SYNTHESIS), the assumptions are weak, and the time complexity of Freya PAGE is optimal when $\sqrt{m} \geq n$.

[a] In Line 3 of their Algorithm 3, they calculate the full gradient, assuming that it can be done for free and not explaining how.
[b] Their convergence rates in Theorems 1 and 3 depend on a bound on the delays $\Delta$, which in turn depends on the performance of the slowest worker. Our method does not depend on the slowest worker if it is too slow (see Section 4.3), which is required for optimality.
[c] We prove better time complexity in Theorem 6, but this result requires the knowledge of $\{\tau_i\}$ in advance, unlike Theorems 7 and 8.

on the relationship between $\tau_1$ and $\tau_n/n$, this could be more efficient or less efficient compared to Hero GD. This shows that the presence of stragglers can eliminate the potential speedup expected from parallelizing the training [Dutta et al., 2018].

**SPIDER/PAGE with 1 worker or $n$ workers and equal data allocation** (Hero PAGE **and** Soviet PAGE)**.** As mentioned in Section 1.2, SPIDER/PAGE can have better *gradient complexity* guarantees than GD. Using the result of Li et al. [2021], the equal data allocation strategy with $n$ workers leads to the time complexity of

$$T_{\text{Soviet PAGE}} := \Theta \left( \tau_n \max \left\{ \frac{m}{n}, 1 \right\} + \tau_n \frac{\delta^0 \max\{L_-, L_\pm\}}{\varepsilon} \max \left\{ \frac{\sqrt{m}}{n}, 1 \right\} \right) \qquad (2)$$

seconds. We refer to this method as Soviet PAGE. In practical regimes, when $\varepsilon$ is small and $L_- \approx L_\pm$, this complexity can be $\sqrt{m}$ better than that of GD. Running PAGE on the fastest worker (which we will call Hero PAGE), we instead get the time complexity $T_{\text{Hero PAGE}} := \Theta \left( \tau_1 m + \tau_1 \delta^0/\varepsilon \sqrt{m} \right)$.

Given these examples, the following question remains unanswered: what is the best possible time complexity in our setting? This paper aims to answer this question.

## 2 Contributions

We consider the finite-sum optimization problem (1) under weak assumptions and develop a new method, Freya PAGE. The method works with arbitrarily heterogeneous and asynchronous computations on the clients without making *any* assumptions about the bounds on the processing times $\tau_i$. We show that the *time complexity* of Freya PAGE is provably better than that of all previously proposed synchronous/asynchronous methods (Table 1). Moreover, we prove a lower bound that guarantees optimality of Freya PAGE in the large-scale regime ($\sqrt{m} \geq n$). The algorithm leverages new computation strategies, ComputeGradient (Alg. 2) and ComputeBatchDifference (Alg. 3), which are generic

---

**Algorithm 1** Freya PAGE

---

1: **Parameters:** starting point $x^0 \in \mathbb{R}^d$, learning rate $\gamma > 0$, minibatch size $S \in \mathbb{N}$, probability
   $p \in (0, 1]$, initialization $g^0 = \nabla f(x^0)$ using ComputeGradient$(x^0)$    (Alg. 2)
2: **for** $k = 0, 1, \ldots, K - 1$ **do**
3:     $x^{k+1} = x^k - \gamma g^k$
4:     Sample $c^k \sim$ Bernoulli$(p)$
5:     **if** $c^k = 1$ **then**                                                                    (with probability $p$)
6:         $\nabla f(x^{k+1}) = $ ComputeGradient$(x^{k+1})$                                         (Alg. 2)
7:         $g^{k+1} = \nabla f(x^{k+1})$
8:     **else**                                                                                     (with probability $1 - p$)
9:         $\frac{1}{S} \sum_{i \in \mathcal{S}^k} \left( \nabla f_i(x^{k+1}) - \nabla f_i(x^k) \right) = $ ComputeBatchDifference$(S, x^{k+1}, x^k)$    (Alg. 3)
10:        $g^{k+1} = g^k + \frac{1}{S} \sum_{i \in \mathcal{S}^k} \left( \nabla f_i(x^{k+1}) - \nabla f_i(x^k) \right)$
11:    **end if**
12: **end for**

(note): $\mathcal{S}^k$ is a set of i.i.d. indices that are sampled from $[m]$, *uniformly with replacement*, $\left| \mathcal{S}^k \right| = S$

---

and can be used in any other asynchronous method. These strategies enable the development of our new SGD method (Freya SGD); see Sections 6 and H. Experiments from Section A on synthetic optimization problems and practical logistic regression tasks support our theoretical results.

## 3   The Design of the New Algorithm

It is clear that to address the challenges arising in the setup under consideration and achieve optimality, a distributed algorithm has to adapt to and effectively utilize the heterogeneous nature of the underlying computational infrastructure. With this in mind, we now present a new algorithm, Freya PAGE, that can efficiently coordinate and synchronize computations across the $n$ devices, accommodating arbitrarily varying processing speeds, while mitigating the impact of slow devices or processing delays on the overall performance of the system.

Freya PAGE is formalized in Algorithm 1. The update rule is just the regular PAGE [Li et al., 2021] update: at each iteration, with some (typically small) probability $p$, the algorithm computes the full gradient $\nabla f(x^{k+1})$, and otherwise, it samples a minibatch $\mathcal{S}^k$ of size $S$ and reuses the gradient estimator $g^k$ from the previous iteration, updated by the cheaper-to-compute adjustment $\frac{1}{S} \sum_{i \in \mathcal{S}^k} \left( \nabla f_i(x^{k+1}) - \nabla f_i(x^k) \right)$.

Within Algorithm 1, at each iteration we call one of two subroutines: ComputeGradient (Alg. 2, performing the low-probability step), and ComputeBatchDifference (Alg. 3, performing the high-probability step). Let us focus on ComputeGradient, designed to collect the full gradient: it takes a point $x$ as input and returns $\nabla f(x) = \frac{1}{m} \sum_{i=1}^{m} \nabla f_i(x)$. There exist many strategies for implementing this calculation, some of which were outlined in Section 1.3. The most naive one is to assign the task of calculating the whole gradient $\nabla f$ to a single worker $i$, resulting in a worst-case running time of $m \tau_i$ seconds for ComputeGradient. Another possible strategy is to distribute the functions $\{f_i\}$ evenly among the workers; in this case, calculating $\nabla f$ takes $\tau_n \max\{m/n, 1\}$ seconds in the worst case.

Clearly, we could do better if we *knew* $\{\tau_i\}$ *in advance*. Indeed, let us allocate to each worker $j$ a number of functions $\{f_i\}$ inversely proportional to $\tau_j$. This strategy is reasonable – the faster the worker, the more gradients it can compute. We can show that such a strategy finds $\nabla f$ in

$$\Theta \left( \min_{j \in [n]} \left( \left( \sum_{i=1}^{j} \frac{1}{\tau_i} \right)^{-1} (m + j) \right) \right) \tag{3}$$

seconds in the worst case (see the proof of Theorem 2). This complexity is better than $m \tau_1$ and $\tau_n \max\{m/n, 1\}$ (Theorem 31). However, this approach comes with two major limitations: i) it requires knowledge of the upper bounds $\{\tau_i\}$, ii) even if we have access to $\{\tau_i\}$, the computation environment can be adversarial: theoretically and practically, it is possible that at the beginning the first worker is the fastest and the last worker is the slowest, but after some time, their performances

| **Algorithm 2** ComputeGradient($x$) | **Algorithm 3** ComputeBatchDifference($S, x, y$) |
|---|---|
| 1: **Input:** point $x \in \mathbb{R}^d$ | 1: **Input:** batch size $S \in \mathbb{N}$, points $x, y \in \mathbb{R}^d$ |
| 2: Init $g = 0 \in \mathbb{R}^d$, set $\mathcal{M} = \emptyset$ | 2: Init $g = 0 \in \mathbb{R}^d$ |
| 3: Broadcast $x$ to all workers | 3: Broadcast $x, y$ to all workers |
| 4: For each worker $i \in [n]$, sample $j$ from $[m]$ uniformly and ask it to calculate $\nabla f_j(x)$ | 4: For each worker, sample $j$ from $[m]$ uniformly and ask it to calculate $\nabla f_j(x) - \nabla f_j(y)$ |
| 5: **while** $\mathcal{M} \neq [m]$ **do** | 5: **for** $i = 1, 2, \ldots, S$ **do** |
| 6:     Wait for $\nabla f_p(x)$ from a worker | 6:     Wait for $\nabla f_p(x) - \nabla f_p(y)$ from a worker |
| 7:     **if** $p \in [m] \backslash \mathcal{M}$ **then** | 7:     $g \leftarrow g + \frac{1}{S}(\nabla f_p(x) - \nabla f_p(y))$ |
| 8:         $g \leftarrow g + \frac{1}{m}\nabla f_p(x)$ | 8:     Sample $j$ from $[m]$ uniformly and ask this worker to calculate $\nabla f_j(x) - \nabla f_j(y)$ |
| 9:         Update $\mathcal{M} \leftarrow \mathcal{M} \cup \{p\}$ | 9: **end for** |
| 10:    **end if** | 10: Return $g$ |
| 11:    Sample $j$ from $[m] \backslash \mathcal{M}$ uniformly and ask this worker to calculate $\nabla f_j(x)$ | |
| 12: **end while** | |
| 13: Return $g = \frac{1}{m} \sum\limits_{i=1}^{m} \nabla f_i(x)$ | |

Notes: i) the workers can aggregate $\nabla f_p$ locally, and the algorithm can call AllReduce once to collect all calculated gradients. ii) By splitting $[m]$ into blocks, instead of one $\nabla f_p$, we can ask the workers to calculate the sum of one block in Alg. 2 (and use a similar idea in Alg. 3).

swap. Consequently, the first worker might end up being assigned the largest batch, despite now having the lowest performance. Thus, this strategy is not robust to time-varying speeds.

**New gradient computation strategy.** The key innovation of this work lies in the introduction of new computation strategies: Algorithms 2 and 3. We start by examining Algorithm 2. It first broadcasts the input $x \in \mathbb{R}^d$ to all workers. Then, for each worker, it samples $j$ *uniformly* from $[m]$ and asks it to calculate $\nabla f_j(x)$ (with a non-zero probability, two workers can be assigned the same computation). Then, the algorithm enters the loop and waits for any worker to finish their calculations. Once this happens, it asks this worker to compute a stochastic gradient with a new index sampled *uniformly* from the set $[m] \backslash \mathcal{M}$ of indices that have not yet been processed (again, it is possible to resample an index previously assigned to another worker). This continues until all indices $i \in [m]$ have been processed and the full gradient $\frac{1}{m} \sum_{i=1}^{m} \nabla f_i$ has been collected. Unlike the previous strategies, our Algorithm 2 does not use $\{\tau_i\}$, thus being *robust and adaptive to the changing compute times*. Furthermore, we can prove that its time complexity (almost) equals (3):

**Theorem 1.** *The expected time needed by Algorithm 2 to calculate $g = \frac{1}{m} \sum\limits_{i=1}^{m} \nabla f_i$ is at most*

$$12 \min_{j \in [n]} \left( \left( \sum_{i=1}^{j} \frac{1}{\tau_i} \right)^{-1} (m + \min\{m, n\} \log(\min\{m, n\}) + j) \right) \tag{4}$$

*seconds.*

The result (4) (the proof of which can be found in Section C) is slightly worse than (3) due to the extra $\min\{m, n\} \log(\min\{m, n\})$ term. This term arises because a worker may be assigned a gradient $\nabla f_j(x)$ that was previously assigned to another worker (in Line 11 of Algorithm 2). Hence, with a *small* (but non-zero) probability, two workers can perform the same calculations. However, typically the number of samples $m$ is much larger than the number of workers $n$. If we assume that $m \geq n \log n$, which is satisfied in many practical scenarios, then the time complexity (4) equals

$$\Theta \left( \min_{j \in [n]} \left( \left( \sum_{i=1}^{j} \frac{1}{\tau_i} \right)^{-1} (m + j) \right) \right)$$

and the term $\min\{m, n\} \log(\min\{m, n\})$ never dominates. Since this complexity is not worse than (3), our strategy behaves as if it knew $\{\tau_i\}$ in advance! To simplify formulas and avoid the logarithmic term, we use the following assumption throughout the main part of this paper:

**Assumption 4.** $m \geq n \log n$, *where $m$ is the # of data samples and $n$ is the # of workers.*

We now proceed to discuss ComputeBatchDifference (Algorithm 3), designed to compute a minibatch of stochastic gradient differences. Both Algorithms 2 and 3 calculate sums. However the latter only

waits until there are $S$ samples in the sum, where some indices in the batch may not be unique. On the other hand, Algorithm 2 must ensure the collection of a full batch of $m$ *unique* stochastic gradients. As a result, Algorithm 3 offers better complexity results and, unlike ComputeGradient, does not suffer from suboptimal guarantees and logarithmic terms, as demonstrated in the theorem below.

**Theorem 2.** *The time needed by Algorithm 3 to calculate $g$ is at most*

$$4 \min_{j \in [n]} \left( \left( \sum_{i=1}^{j} \frac{1}{\tau_i} \right)^{-1} (S + j) \right) \tag{5}$$

*seconds.*

Algorithm 1 uses *uniform sampling with replacement*, implemented in Algorithm 3. However, one can modify the two algorithms slightly to support virtually *any unbiased sampling* (see Section E).

**Note on asynchronicity.**    It is clear that to eliminate the need of waiting for very slow machines, some level of asynchronicity has to be injected into an algorithm for it to be efficient. Asynchronous SGD [Recht et al., 2011, Nguyen et al., 2018, Koloskova et al., 2022, Mishchenko et al., 2022] takes this concept to the extreme by never synchronizing and continually overwriting the updates. Consequently, the algorithm's time complexity is suboptimal. Conversely, imposing limitations on asynchronicity leads to optimal methods, both in our context (in the large-scale regime) and in the scenario examined by Tyurin and Richtárik [2023]. Freya PAGE seamlessly combines synchrony and asynchrony, getting the best out of the two worlds.

## 4   Time Complexities and Convergence Rates

Formulas (3) and (4) will be used frequently throughout the paper. To lighten up the heavy notation, let us define the following mapping.

**Definition 3** (Equilibrium time)**.** A mapping $t^* : \mathbb{R}_{\geq 0} \times \mathbb{R}_{\geq 0}^n \to \mathbb{R}_{\geq 0}$ defined by

$$t^*(S, [\bar{\tau}_i]_{i=1}^n) := \min_{j \in [n]} \left( \left( \sum_{i=1}^{j} \frac{1}{\bar{\tau}_i} \right)^{-1} (S + j) \right) \in [0, \infty] \tag{6}$$

is called the *equilibrium time*. We let $t^*(S) \equiv t^*(S, [\tau_i]_{i=1}^n)$ when considering $\{\tau_i\}$ from Section 1.

Returning to the algorithm, we guarantee the following iteration complexity.

**Theorem 4** (Iteration complexity)**.** *Let Assumptions 1, 2 and 3 hold. Consider any minibatch size $S \in \mathbb{N} := \{1, 2, \ldots\}$, any probability $p \in (0, 1]$, and let the stepsize be $\gamma = \left( L_- + L_\pm \sqrt{\frac{1-p}{pS}} \right)^{-1}$. Then, after*

$$K \geq K_{\mathsf{PAGE}} := \frac{2\delta^0}{\varepsilon} \left( L_- + L_\pm \sqrt{\frac{1-p}{pS}} \right) \tag{7}$$

*iterations of Algorithm 1, we have $\mathbb{E}\left[ \left\| \nabla f(\hat{x}^K) \right\|^2 \right] \leq \varepsilon$, where $\hat{x}^K$ is sampled uniformly at random from the iterates $\{x^0, \ldots, x^{K-1}\}$.*

Theorem 4 states that the iteration complexity is the same as in the optimal PAGE method [Li et al., 2021]. Note that we can guarantee convergence even if the upper bounds $\{\tau_i\}$ are unknown or infinite (as long as there exists some worker that can complete computations within a finite time).

We now derive time complexity guarantees. With probability $p$, the workers need to supply to the algorithm $m$ stochastic gradients at each of the $m$ data samples, which by Theorem 1 can be done in at most $t^*(m)$ seconds (up to a log factor). Otherwise, they compute $S$ differences of stochastic gradients, which by Theorem 2 takes at most $t^*(S)$ seconds (up to a constant factor). The resulting time complexity is given in the theorem below.

**Theorem 5** (Time complexity with free parameters $p$ and $S$)**.** *Consider the assumptions and the parameters from Theorem 4, plus Assumption 4. The expected time complexity of Algorithm 1 is at most*

$$\begin{aligned}
T(p, S, [\tau_i]_{i=1}^n) := {} & 12 \cdot t^*(m, [\tau_i]_{i=1}^n) \\
& + \frac{48\delta^0}{\varepsilon} \left( L_- + L_\pm \sqrt{\frac{1-p}{pS}} \right) \times \left\{ p \cdot t^*(m, [\tau_i]_{i=1}^n) + (1-p) \cdot t^*(S, [\tau_i]_{i=1}^n) \right\}.
\end{aligned} \tag{8}$$

The first term comes from the preprocessing step, where the full gradient is calculated to obtain $g^0 = \nabla f(x^0)$. Here, we use Assumption 4 that $m \geq n \log n$. The result (8) is valid even without this assumption, but at the cost of extra logarithmic factors.

## 4.1 Optimal parameters $S$ and $p$

The time complexity (8) depends on two free parameters, $S \in \mathbb{N}$ and $p \in (0, 1]$. The result below (following from Theorems 13 and 14) determines their optimal choice.

**Theorem 6** (Main result). *Consider the assumptions and parameters from Theorem 4, plus Assumption 4. Up to a constant factor, the time complexity (8) is at least $T([\tau_i]_{i=1}^n) := t^*(m, [\tau_i]_{i=1}^n)$*

$$+ \frac{\delta^0}{\varepsilon} \min \left\{ L_- t^*(m, [\tau_i]_{i=1}^n), \min_{S \in \mathbb{N}} \underbrace{\left[ L_- t^*(S, [\tau_i]_{i=1}^n) + L_\pm \frac{\sqrt{t^*(m,[\tau_i]_{i=1}^n) t^*(S,[\tau_i]_{i=1}^n)}}{\sqrt{S}} \right]}_{F(S):=} \right\}, \quad (9)$$

*and this lower bound is achieved with $S^* = \arg\min\limits_{S \in \mathbb{N}} F(S)$ and $p^* = p^*(S^*)$, where*

$$p^*(S) = \begin{cases} 1, & \text{if } L_- t^*(m, [\tau_i]_{i=1}^n) \leq L_- t^*(S, [\tau_i]_{i=1}^n) + L_\pm \frac{\sqrt{t^*(m,[\tau_i]_{i=1}^n) t^*(S,[\tau_i]_{i=1}^n)}}{\sqrt{S}}, \\ \frac{t^*(S,[\tau_i]_{i=1}^n)}{t^*(m,[\tau_i]_{i=1}^n)}, & \text{otherwise.} \end{cases}$$

Result (9) is the best possible time complexity that can be achieved with the Freya PAGE method. Unfortunately, the final time complexity has non-trivial structure, and the optimal parameters depend on $\{\tau_i\}$ in the general case. If we have access to all parameters and times $\{\tau_i\}$, then (9), $S^*$, and $p^*$ can be computed efficiently. Indeed, the main problem is to find $\min_{S \in \mathbb{N}} F(S)$, which can be solved, for instance, by using the bisection method, because $L_- t^*(S, [\tau_i]_{i=1}^n)$ is non-decreasing and $L_\pm \sqrt{t^*(m, [\tau_i]_{i=1}^n) t^*(S, [\tau_i]_{i=1}^n)}/\sqrt{S}$ is non-increasing in $S$.

## 4.2 Optimal parameters $S$ and $p$ in the large-scale regime

Surprisingly, we can significantly simplify the choice of the optimal parameters $S$ and $p$ in the large-scale regime, when $\sqrt{m} \geq n$. This is a weak assumption, since typically the number of data samples is much larger than the number of workers.

**Theorem 7** (Main result in the large-scale regime). *Consider the assumptions and parameters from Theorem 4, plus Assumption 4. Up to a constant factor and smoothness constants, if $\sqrt{m} \geq n$, then the optimal choice of parameters in (8) is $S^* = \lceil \sqrt{m} \rceil$ and $p^* = 1/\sqrt{m}$. For this choice, the expected time complexity of Algorithm 1 is at most*

$$T(1/\sqrt{m}, \sqrt{m}, [\tau_i]_{i=1}^n) = 12 t^*(m, [\tau_i]_{i=1}^n) + \frac{192 \delta^0 \max\{L_-, L_\pm\}}{\varepsilon} t^*(\sqrt{m}, [\tau_i]_{i=1}^n) \quad (10)$$

*seconds. The iteration complexity with $S = \lceil \sqrt{m} \rceil$ and $p = 1/\sqrt{m}$ is $K_{\mathsf{PAGE}} \leq 4\delta^0 \max\{L_-, L_\pm\}/\varepsilon$.*

We cannot guarantee that $S = \lceil \sqrt{m} \rceil$ and $p = 1/\sqrt{m}$ is the *optimal* pair when $\sqrt{m} < n$, but it is a valid choice for all $m \geq 1$. Note that (10) is true if $m \geq n \log n$, and it is true up to a log factor if $m < n \log n$. In light of Theorem 15, we can further refine Theorem 7 if the ratio $L_\pm/L$ is known:

**Theorem 8** (Main result in the large-scale regime using the ratio $L_\pm/L$). *Consider the assumptions and parameters from Theorem 4, plus Assumption 4. The expected time complexity of Algorithm 1 is at most $\Theta(t^*(m, [\tau_i]_{i=1}^n) + \delta^0 L_-/\varepsilon \times t^*(\min\{\max\{\lceil L_\pm \sqrt{m}/L_- \rceil, 1\}, m\}, [\tau_i]_{i=1}^n))$ seconds, where $S = \min\{\max\{\lceil L_\pm \sqrt{m}/L_- \rceil, 1\}, m\}$ and $p = S/m$.*

For brevity reasons, we will continue working with the result from Theorem 7 in the main part of this paper. Note that the optimal parameters do not depend on $\{\tau_i\}$, and can be easily calculated since the number of functions $m$ is known in advance. Hence, our method is *fully adaptive to changing and heterogeneous compute times of the workers*.

Even if the bounds are unknown and $\tau_i = \infty$ for all $i \in [n]$, our method converges after $4\delta^0 \max\{L_-, L_\pm\}/\varepsilon$ iterations, and calculates the optimal number of stochastic gradients equal to $\mathcal{O}(m + \sqrt{m}\delta^0 \max\{L_-, L_\pm\}/\varepsilon)$.

## 4.3 Discussion of the time complexity

Let us use Definition 3 and unpack the second term in the complexity (10), namely,

$$\frac{\delta^0 \max\{L_-, L_\pm\}}{\varepsilon} \min_{j \in [n]} \left( \left( \sum_{i=1}^{j} \frac{1}{\tau_i} \right)^{-1} (\sqrt{m} + j) \right).$$

A somewhat similar expression involving $\min_{j \in [n]}$ and harmonic means was obtained by Tyurin and Richtárik [2023], Tyurin et al. [2024] for minimizing expectation under the bounded variance assumption. The term $\delta^0 \max\{L_-, L_\pm\}/\varepsilon$ is standard in optimization [Nesterov, 2018, Lan, 2020] and describes the difficulty of the problem (1). The term $\min_{j \in [n]}((\sum_{i=1}^{j} 1/\tau_i)^{-1}(\sqrt{m} + j))$ represents the average time of one iteration and has some nice properties. For instance, if the last worker is slow and $\tau_n \approx \infty$, then $\min_{j \in [n]}(\cdots) = \min_{j \in [n-1]}(\cdots)$, so the complexity effectively ignores it. Moreover, if $j^*$ is an index that minimizes $\min_{j \in [n]}(\cdots)$, then $\min_{j \in [n]}((\sum_{i=1}^{j} 1/\tau_i)^{-1}(\sqrt{m} + j)) = ((\sum_{i=1}^{j^*} 1/\tau_i)^{-1}(\sqrt{m} + j^*))$. The last formula, again, does not depend on the slowest workers $\{j^* + 1, \ldots, n\}$, which are automatically excluded from the time complexity expression. The same reasoning applies to the term $t^*(m, [\tau_i]_{i=1}^{n})$. Let us now consider some extreme examples which are meant to shed some light on our time complexity result (10):

**Example 1.** *[Equally Fast Workers] Suppose that the upper bounds on the processing times are equal, i.e., $\tau_j = \tau$ for all $j \in [n]$. Then*

$$T(1/\sqrt{m}, \sqrt{m}, [\tau_i]_{i=1}^{n}) = \Theta \left( \tau \max\left\{ \frac{m}{n}, 1 \right\} + \tau \frac{\delta^0 \max\{L_-, L_\pm\}}{\varepsilon} \max\left\{ \frac{\sqrt{m}}{n}, 1 \right\} \right).$$

The complexity in Example 1 matches that in (2), which makes sense since Soviet PAGE is a reasonable method when $\{\tau_i\}$ are equal. Note that the reduction happens without prior knowledge of $\{\tau_i\}$.

**Example 2.** *[Infinitely Fast Worker] If $\tau_1 = 0$, then $T(1/\sqrt{m}, \sqrt{m}, [\tau_i]_{i=1}^{n}) = 0$.*

**Example 3.** *[Infinitely Slow Workers] If $\tau_j = \infty \, \forall j \in [n]$, then $T(1/\sqrt{m}, \sqrt{m}, [\tau_i]_{i=1}^{n}) = \infty$.*

**Example 4.** *[Extremely Slow Workers] Suppose that the times $\tau_j < \infty$ are fixed $\forall j \leq j_B$ and $\tau_j \geq B \, \forall j > j_B$ for some $B$ large enough. Then $T(1/\sqrt{m}, \sqrt{m}, [\tau_i]_{i=1}^{n}) = T(1/\sqrt{m}, \sqrt{m}, [\tau_i]_{i=1}^{j_B})$.*

Example 4 says that the workers whose processing time is too large are *ignored*, which supports the discussion preceding the examples.

## 4.4 Dynamic bounds

It turns out that the guarantees from Theorem 7 can be generalized to the situation where the compute times $\{\tau_i\}$ are allowed to dynamically change throughout the iterations. Consider the $k^{\text{th}}$ iteration of Algorithm 1 and assume that worker $i$ calculates one $\nabla f_j$ in at most $\tau_i^k \in [0, \infty]$ seconds $\forall i \in [n], j \in [m]$. Clearly, $\max_{k \geq -1} \tau_i^k \leq \tau_i \, \forall i \in [n]$ (where $\{\tau_i^{-1}\}$ are upper bounds from the preprocessing step), but $\tau_i^k$ can be arbitrarily smaller than $\tau_i$ (possibly $\tau_i^k = 0$ and $\tau_i = \infty$).

**Theorem 9.** *Consider the assumptions and parameters from Theorem 4, plus Assumption 4. Up to a constant factor, the expected time complexity of Algorithm 1 with $S = \lceil \sqrt{m} \rceil$ and $p = 1/\sqrt{m}$ is at most*

$$t^*(m, [\tau_{\pi_{-1,i}}^{-1}]_{i=1}^{n}) + \sum_{k=0}^{\lceil 4\delta^0 \max\{L_-, L_\pm\}/\varepsilon \rceil} t^*(\sqrt{m}, [\tau_{\pi_{k,i}}^{k}]_{i=1}^{n}), \tag{11}$$

*where $\pi_{k,\cdot}$ is a permutation such that $\tau_{\pi_{k,1}}^{k} \leq \cdots \leq \tau_{\pi_{k,n}}^{k}$ for all $k \geq -1$. (This theorem follows from Theorem 22 with the chosen parameters).*

Hence, our algorithm is *adaptive to the dynamic compute times* $\{\tau_i^k\}$. Let us consider an example with 2 workers. Assume that the first worker is stable: $\tau_1^k = \tau$ for all $k \geq 0$, and the second worker is unstable: $\tau_2^0 = \tau$ is small in the first iteration, and $\tau_2^1 \approx \infty$ in the second iteration. For explanation purposes, we ignore the preprocessing term $t^*(m, \cdot)$, which is not a factor if $\varepsilon$ is small. Then,

$$(11) = t^*(\sqrt{m}, [\tau_{\pi_{0,1}}^{0}, \tau_{\pi_{0,2}}^{0}]) + t^*(\sqrt{m}, [\tau_{\pi_{1,1}}^{1}, \tau_{\pi_{1,2}}^{1}]) + \ldots = t^*(\sqrt{m}, [\tau, \tau]) + t^*(\sqrt{m}, [\tau]) + \ldots$$

because $t^*(\sqrt{m}, [\tau^1_{\pi_{1,1}}, \tau^1_{\pi_{1,2}}]) = t^*(\sqrt{m}, [\tau])$ when $\tau^1_2 \approx \infty$. The time complexity in the second iteration depends on the first (stable) worker only, which is reasonable since $\tau^1_2 \approx \infty$, and this happens *automatically*. At the same time, the first term $t^*(\sqrt{m}, [\tau, \tau])$ depends on both workers, and this iteration will be faster because $t^*(\sqrt{m}, [\tau, \tau]) \leq t^*(\sqrt{m}, [\tau])$.

## 4.5 Comparison with previous strategies from Section 1.3

Our time complexities (9) and (10) are better than all known previous guarantees if $m \geq n \log n$. In particular, $T(1/\sqrt{m}, \sqrt{m}, [\tau_i]_{i=1}^n) \leq T_{\text{Soviet PAGE}}$ (from (2)), because $t^*(\sqrt{m}, [\tau_i]_{i=1}^n) \lesssim \sqrt{m}\tau_n/n$ (Theorem 31). In fact, since $\lim_{\tau_n \to \infty} t^*(\sqrt{m}, [\tau_i]_{i=1}^n) = t^*(\sqrt{m}, [\tau_i]_{i=1}^{n-1}) < \infty$ and $\lim_{\tau_n \to \infty} = \sqrt{m}\tau_n/n = \infty$, $T_{\text{Soviet PAGE}}$ can be arbitrarily larger. We also improve on Hero PAGE (see Remark 32).

## 4.6 Comparison with existing asynchronous variance reduced methods

Several studies have explored asynchronous variance reduced algorithms. Essentially all of them are variants of the existing synchronous methods discussed in Section 1.2 and depend on the slowest worker in every iteration. There have been several attempts to combine variance reduction techniques with asynchronous computations. Perhaps the most relevant baseline is SYNTHESIS, an asynchronous variant of SPIDER [Fang et al., 2018] introduced by Liu et al. [2022]. The obtained *gradient complexity* matches that of SPIDER in terms of dependence on $m$, but scales linearly with the bound on the time performance of the slowest worker, making it non-adaptive to slow computations. Moreover, in Line 3 of their Algorithm 3, the full gradient is calculated, assuming that it can be done for free. Lastly, the analysis assumes the gradients to be bounded.

# 5 Lower Bound

In previous sections, we showed that Freya PAGE converges in at most (9) or (10) seconds, providing better time complexity guarantees compared to all previous methods. The natural question is: how good are these time complexities, and can they be improved? In Section J, we formalize our setup and prove Theorems 29 and 30, which collectively yield the following lower bound.

**Theorem 10** (Less formal version of Theorems 29 and 30). *Assume that $0 < \tau_1 \leq \cdots \leq \tau_n$ and take any $L_+, \delta^0, \varepsilon > 0$ and $m \in \mathbb{N}$ such that $\varepsilon < c_1 L_+ \delta^0$. Then, for any (zero-respecting) algorithm, there exists a function $f$ that satisfies $f(0) - f^* \leq \delta^0$ and Assumption 2, such that it is impossible to find an $\varepsilon$–stationary point faster than in*

$$\Omega\left(t^*(m, [\tau_i]_{i=1}^n) + \frac{\delta^0 L_+}{\sqrt{m}\varepsilon} t^*(m, [\tau_i]_{i=1}^n)\right) \tag{12}$$

*seconds using uniform sampling with replacement.*

Comparing (10) and (12), we see that Freya PAGE is *optimal* under Assumptions 1 and 2 in the large-scale regime ($\sqrt{m} \geq n$). Indeed, without Assumption 3, we have $\max\{L_-, L_\pm\} = \max\{L_-, L_+\} = L_+$. Up to constant factor, (10) is less or equal to (12) since

$$t^*(\sqrt{m}, [\tau_i]_{i=1}^n) = \min_{j \in [n]}\left(\left(\sum_{i=1}^j \frac{1}{\tau_i}\right)^{-1}(\sqrt{m}+j)\right) = \frac{1}{\sqrt{m}}\min_{j \in [n]}\left(\left(\sum_{i=1}^j \frac{1}{\tau_i}\right)^{-1}(m+\sqrt{m}j)\right)$$

$$\overset{\sqrt{m}\geq n}{\leq} \frac{2}{\sqrt{m}}\min_{j \in [n]}\left(\left(\sum_{i=1}^j \frac{1}{\tau_i}\right)^{-1}(m+j)\right) = \frac{2}{\sqrt{m}}t^*(m, [\tau_i]_{i=1}^n).$$

This is *the first optimal method for the problem we consider*, and Theorem 10 gives *the first lower bound*.

# 6 Using the Developed Strategies in Other Methods

ComputeBatchDifference (Algorithm 3) is a generic subroutine and can be used in other methods. In Section H, we introduce Freya SGD, a simple algorithm with update rule $x^{k+1} = x^k - \gamma \times \text{ComputeBatch}(S, x^k)$, where $S$ is a batch size and ComputeBatch (Algorithm 4) is a

minor modification of ComputeBatchDifference. Theorem 25 establishes that Freya SGD converges in $\mathcal{O}\left(1/\varepsilon \times t^*\left(1/\varepsilon, [\tau_i]_{i=1}^n\right)\right)$ seconds (where we only keep the dependence on $\varepsilon$ and $\{\tau_i\}$). For small $\varepsilon$, this complexity is worse than (10), but it can be better, for instance, in the interpolation regime [Schmidt and Roux, 2013, Ma et al., 2018]. Freya SGD resembles Rennala SGD [Tyurin and Richtárik, 2023], but unlike the latter, it is specialized to work with finite-sum problems (1) and does not require the $\sigma^2$–bounded variance assumption on stochastic gradients (which is not satisfied in our setting).

## Acknowledgments and Disclosure of Funding

The research reported in this publication was supported by funding from King Abdullah University of Science and Technology (KAUST): i) KAUST Baseline Research Scheme, ii) Center of Excellence for Generative AI, under award number 5940, iii) SDAIA-KAUST Center of Excellence in Artificial Intelligence and Data Science. The work of A.T. was partially supported by the Analytical center under the RF Government (subsidy agreement 000000D730321P5Q0002, Grant No. 70-2021-00145 02.11.2021).

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

# APPENDIX

# Contents

# A Experiments

We compare Freya PAGE with Rennala SGD, Asynchronous SGD, and Soviet PAGE on nonconvex quadratic optimization tasks and practical machine learning problems. The experiments were conducted in Python 3.8 with Intel(R) Xeon(R) Gold 6248 CPU @ 2.50GHz. We developed a library that emulates the working behavior of thousands of nodes.

## A.1 Experiments with nonconvex quadratic optimization

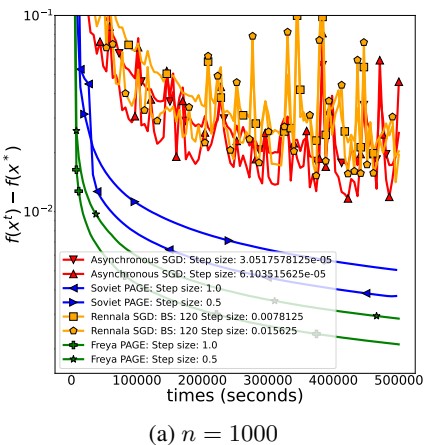
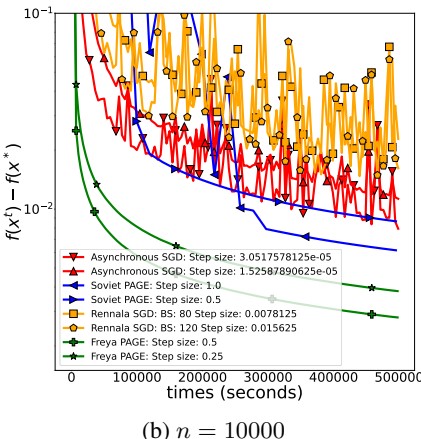

(a) $n = 1000$        (b) $n = 10000$

Figure 1: Experiments with nonconvex quadratic optimization tasks. We plot function suboptimality against elapsed time.

In the first set of experiments, we compare the algorithms on a synthetic quadratic optimization task generated using the procedure from Section I. To ensure robust and fair comparison, we fix the performance of each worker and emulate our setup by assuming that the $i^{\text{th}}$ worker requires $\sqrt{i}$ seconds to calculate a stochastic gradient. For each algorithm, we fine-tune the step size from the set $\{2^i \mid i \in [-20, 20]\}$. Uniform sampling with replacement is used across all methods. In Freya PAGE, we set $S = \lceil \sqrt{m} \rceil$ according to Theorem 7. We consider $n \in \{1000, 10000\}$ and in each case plot the best run of each method.

The results are presented in Figure 1. It is evident that our new method, Freya PAGE, has the best convergence performance among all algorithms considered. The convergence behavior of Rennala SGD and Asynchronous SGD is very noisy, and both achieve lower accuracy than Freya PAGE. Furthermore, the gap between Freya PAGE and Soviet PAGE widens with increasing $n$ because Soviet PAGE is not robust to the presence of slow workers.

## A.2 Experiments with logistic regression

Table 2: Mean and variance of algorithm accuracies on the MNIST test set during the final 100K seconds of the experiments from Figure 2b.

| Method | Accuracy | Variance of Accuracy |
|---|---|---|
| Asynchronous SGD [Koloskova et al., 2022] [Mishchenko et al., 2022] | 92.60 | 5.85e-07 |
| Soviet PAGE [Li et al., 2021] | 92.31 | 1.62e-07 |
| Rennala SGD [Tyurin and Richtárik, 2023] | 92.37 | 3.12e-06 |
| **Freya PAGE** | **92.66** | **1.01e-07** |

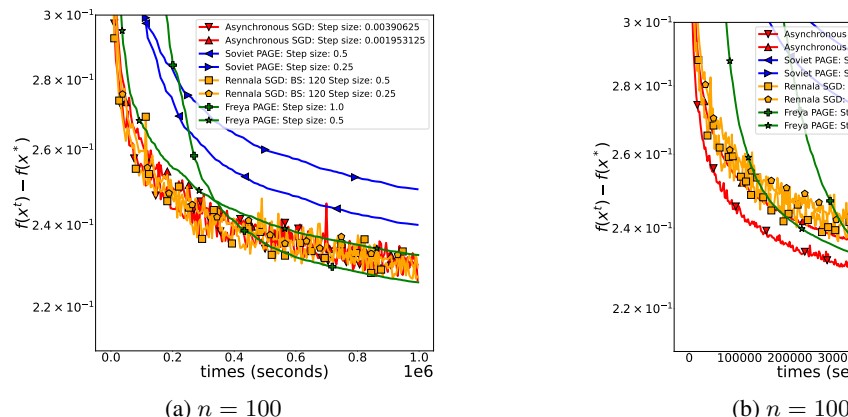

(a) $n = 100$        (b) $n = 10000$

Figure 2: Experiments with the logistic regression problem on the `MNIST` dataset.

We now consider the logistic regression problem on the `MNIST` dataset [LeCun et al., 2010], where each algorithm samples one data point at a time. The results of the experiments are presented in Figure 2. The difference between Freya PAGE and Rennala SGD/Asynchronous SGD is not as pronounced as in Section A.1: the methods have almost the same performance for this particular problem. However, our method still outperforms its competitors in the low accuracy regime, and is significantly better than Soviet PAGE.

A critical disadvantage of Rennala SGD and Asynchronous SGD is their noisy behavior, evident in both the plots and reflected in higher variance of the accuracy (see Table 2). In contrast, the iterations of Freya PAGE in Figure 2 are smooth, and its accuracy exhibits the lowest variance, as shown in Table 2. This stability can be attributed to the variance-reduction nature of Freya PAGE.

# B The Time Complexity Guarantees of Algorithms 3 and 4

In addition to ComputeBatchDifference (Algorithm 3) introduced in the main part, we also analyze ComputeBatch (Algorithm 4) that is similar to ComputeBatchDifference, but calculates a minibatch of stochastic gradients $\nabla f_i(x)$ instead of stochastic gradient differences $\nabla f_i(x) - \nabla f_i(y)$.

---

**Algorithm 4** ComputeBatch($S, x$)

---

1: **Input:** batch size $S \in \mathbb{N}$, point $x \in \mathbb{R}^d$
2: Init $g = 0 \in \mathbb{R}^d$
3: Broadcast $x$ to all workers
4: For each worker, sample $j$ from $[m]$ (uniformly) and ask it to calculate $\nabla f_j(x)$
5: **for** $i = 1, 2, \ldots, S$ **do**
6:     Wait for $\nabla f_p(x)$ from a worker
7:     $g \leftarrow g + \frac{1}{S}\nabla f_p(x)$
8:     Sample $j$ from $[m]$ (uniformly) and ask this worker to calculate $\nabla f_j(x)$
9: **end for**
10: Return $g$

---

**Theorem 2.** *The time needed by Algorithm 3 to calculate $g$ is at most*

$$4 \min_{j \in [n]} \left( \left( \sum_{i=1}^{j} \frac{1}{\tau_i} \right)^{-1} (S + j) \right) \tag{5}$$

*seconds.*

*Proof.* Let

$$t = 4 \min_{j \in [n]} \left( \left( \sum_{i=1}^{j} \frac{1}{\tau_i} \right)^{-1} (S + j) \right).$$

As soon as some worker finishes calculating the stochastic gradient difference, it immediately starts computing the difference of the next pair. Hence, by the time $t$, all workers will have processed at least

$$\sum_{i=1}^{n} \left\lfloor \frac{t}{2\tau_i} \right\rfloor$$

pairs. Assume that

$$j^* = \arg \min_{j \in [n]} \left( \left( \sum_{i=1}^{j} \frac{1}{\tau_i} \right)^{-1} (S + j) \right).$$

Using Lemma 4, we have $t \geq 4\tau_i$ for all $i \leq j^*$. Thus, we get $\frac{t}{2\tau_i} \geq 1$ for all $i \leq j^*$ and

$$\sum_{i=1}^{n} \left\lfloor \frac{t}{2\tau_i} \right\rfloor \geq \sum_{i=1}^{j^*} \left\lfloor \frac{t}{2\tau_i} \right\rfloor \geq \sum_{i=1}^{j^*} \frac{t}{4\tau_i}$$

$$= \frac{1}{4} \left( \sum_{i=1}^{j^*} \frac{1}{\tau_i} \right) \left( 4 \left( \sum_{i=1}^{j^*} \frac{1}{\tau_i} \right)^{-1} (S + j^*) \right) = S + j^* \geq S.$$

We can conclude that by the time (5), the algorithm will have calculated $S$ pairs of stochastic gradients and exited the loop. $\square$

**Theorem 11.** *The time needed by Algorithm 4 to calculate $g$ is at most*

$$2 \min_{j \in [n]} \left( \left( \sum_{i=1}^{j} \frac{1}{\tau_i} \right)^{-1} (S + j) \right) \tag{13}$$

*seconds.*

*Proof.* The proof of this theorem is essentially the same as the proof of Theorem 2. The only difference is that Algorithm 4 calculates $\nabla f_i(x)$ instead of $\nabla f_i(x) - \nabla f_i(y)$. □

## C  The Time Complexity Guarantees of Algorithms 2 and 5

Instead of Algorithm 2, we analyze a more general Algorithm 5 that reduces to Algorithm 2 when $\mathcal{S} = [m]$.

---

**Algorithm 5** ComputeBatchAnySampling($\mathcal{S}, x$)

---

1: **Input:** multiset $\mathcal{S}$, point $x \in \mathbb{R}^d$
2: Init $g = 0 \in \mathbb{R}^d$, multiset $\mathcal{M} = \emptyset$
3: Broadcast $x$ to all workers
4: For each worker, sample $j$ from $\mathcal{S}$ (uniformly) and ask it to calculate $\nabla f_j(x)$
5: **while** $\mathcal{M} \neq \mathcal{S}$ **do**
6:     Wait for $\nabla f_p(x)$ from a worker
7:     **if** $p \in \mathcal{S} \backslash \mathcal{M}$ **then**
8:         $g \leftarrow g + \frac{1}{|\mathcal{S}|} \nabla f_p(x)$
9:         Update $\mathcal{M} \leftarrow \mathcal{M} \cup \{p\}$
10:     **end if**
11:     Sample $j$ from $\mathcal{S} \backslash \mathcal{M}$ (uniformly) and ask this worker to calculate $\nabla f_j(x)$
12: **end while**
13: Return $g = \frac{1}{|\mathcal{S}|} \sum_{i \in \mathcal{S}} \nabla f_i(x)$

---

**Theorem 12.** *The expected time needed by Algorithm 5 to calculate $g = \frac{1}{|\mathcal{S}|} \sum_{i \in \mathcal{S}} \nabla f_i$ is at most*

$$12 \min_{j \in [n]} \left( \left( \sum_{i=1}^{j} \frac{1}{\tau_i} \right)^{-1} (|\mathcal{S}| + \min\{|\mathcal{S}|, n\} \log(\min\{|\mathcal{S}|, n\}) + j) \right) \tag{14}$$

*seconds.*

*Proof Sketch:* While the following proof is technical, the intuition and idea behind it and the algorithm are relatively simple. For simplicity, assume that $n \geq |\mathcal{S}|$. The set $\mathcal{S} \backslash \mathcal{M}$ includes all indices that have not yet been calculated. Each worker is assigned a new random index from $\mathcal{S} \backslash \mathcal{M}$ and starts the calculation of the gradient. At the beginning of the algorithm, when the set $\mathcal{S} \backslash \mathcal{M}$ is large, the probability that two workers are assigned the same index is very small. Hence, using the same idea as in the proof of Theorem 2, the workers will calculate $\approx |\mathcal{S}| - n$ stochastic gradients after

$$\approx \min_{j \in [n]} \left( \left( \sum_{i=1}^{j} \frac{1}{\tau_i} \right)^{-1} (|\mathcal{S}| - n + j) \right)$$

seconds. However, once the size of $\mathcal{S} \backslash \mathcal{M}$ becomes roughly equal to $n$, the probability that two workers sample the same index increases. In the final steps of the algorithm, we encounter the same issue as in the famous *coupon collector's problem*, resulting in an additional factor of $n \log n$ because some stochastic gradients will be calculated multiple times.

*Proof.* Let us define $S = |\mathcal{S}|$ and take any $k \in [n]$. We refer to the workers with the upper bounds $\tau_i$ such that $\tau_i \leq \tau_k$ as "fast", and the others will be termed "slow".

Consider the moment when the algorithm samples $j$ from the set $\mathcal{S} \backslash \mathcal{M}$ to allocate it to one of the "fast" workers (Line 4 or 11). The probability of sampling $j$ such that $\nabla f_j(x)$ is currently being calculated by another "fast" worker is

$$\frac{|\{\text{indices from } \mathcal{S} \backslash \mathcal{M} \text{ taken by "fast" workers}\}|}{|\{\text{indices from } \mathcal{S} \backslash \mathcal{M} \text{ taken by "fast" workers}\}| + |\{\text{indices from } \mathcal{S} \backslash \mathcal{M} \text{ not taken by "fast" workers}\}|}$$

$$\leq \frac{\min\{k, S\}}{\min\{k, S\} + |\{\text{indices from } \mathcal{S}\backslash\mathcal{M} \text{ not taken by "fast" workers}\}|}$$

because there are at most $k$ "fast" workers and at most $S$ distinct stochastic gradients. Let us define the set

$$\mathcal{U} := \{\text{indices from } \mathcal{S}\backslash\mathcal{M} \text{ not taken by "fast" workers}\}.$$

A "fast" worker can be "unlucky" and start calculating a stochastic gradient that is being computed by another "fast" worker. However, with probability at least

$$\geq \frac{|\mathcal{U}|}{\min\{k, S\} + |\mathcal{U}|},$$

it will take a new index $j$ that was not previously taken by another "fast" worker.

Thus, the while loop of the algorithm defines a Markov process that begins with some $\mathcal{U} \subseteq \mathcal{S}$. The size of $\mathcal{U}$ decreases by one with probability at least $\frac{|\mathcal{U}|}{\min\{k,S\}+|\mathcal{U}|}$ in iterations where the algorithm samples $j$ from $\mathcal{U}$ and asks a "fast" worker to calculate the stochastic gradient. Additionally, the size of $\mathcal{U}$ can decrease by one when a "slow" worker finishes calculating a stochastic gradient from $\mathcal{U}$.

Let $\bar{t}$ be the time required for the Markov process to reach the state $\mathcal{U} = \emptyset$. Then, the while loop in Algorithm 2 will finish after at most

$$\bar{T} := \bar{t} + \tau_k \tag{15}$$

seconds because once $\mathcal{U} = \emptyset$, all non-processed indices from $\mathcal{S}\backslash\mathcal{M}$ are assigned to the "fast" workers, so calculating the remaining stochastic gradients will take at most $\tau_k$ seconds.

It remains to estimate $\bar{t}$. Let $\eta_p$ be the number of iterations of the while loop where the algorithm samples $j$ from $\mathcal{S}\backslash\mathcal{M}$ and asks a "fast" worker to calculate the stochastic gradient when $|\mathcal{U}| = p$. By the definition of the Markov chain, we have

$$\mathbb{E}\left[\eta_p\right] \leq \left(\frac{p}{\min\{k, S\} + p}\right)^{-1} = 1 + \frac{\min\{k, S\}}{p} \tag{16}$$

because with probability at least $\frac{p}{\min\{k,S\}+p}$, one of the ("lucky") "fast" workers receives $j$ from $\mathcal{U}$ and decreases the size of $\mathcal{U}$ by $1$ ($\eta_p$ has a geometric-like distribution).

Since $|\mathcal{U}| \leq S$ at the beginning of the while loop, it is sufficient for the "fast" workers to calculate at most

$$\sum_{p=1}^{S} (\eta_p + 1)$$

stochastic gradients to ensure that $\mathcal{U} = \emptyset$ (it is possible that some stochastic gradients will be calculated many times). Indeed, if $|\mathcal{U}| = p$ for the first moment, then after $\eta_p + 1$ calculations of stochastic gradients by the "fast" workers, the size of set will be at most $p - 1$. The last "plus one" calculation can only happen when $|\mathcal{U}| = p - 1$.

The time required for the "fast" workers to process this number of stochastic gradients is at most

$$t' = 2\left(\sum_{i=1}^{k} \frac{1}{\tau_i}\right)^{-1} \left(\sum_{p=1}^{S}(\eta_p + 1)\right) + \tau_k,$$

because for this choice of $t'$, we have

$$\sum_{i=1}^{k} \left\lfloor \frac{t'}{\tau_i} \right\rfloor \geq \frac{1}{2} \sum_{i=1}^{k} \frac{t'}{\tau_i} \geq \sum_{p=1}^{S}(\eta_p + 1),$$

where $\left\lfloor \frac{t'}{\tau_i} \right\rfloor$ is the number of stochastic gradients that worker $i$ can calculate in $t'$ seconds. Taking expectation gives

$$\mathbb{E}\left[t'\right] = 2\left(\sum_{i=1}^{k} \frac{1}{\tau_i}\right)^{-1} \left(\sum_{p=1}^{S} \mathbb{E}\left[\eta_p + 1\right]\right) + \tau_k$$

$$\stackrel{(16)}{\leq} 2 \left( \sum_{i=1}^{k} \frac{1}{\tau_i} \right)^{-1} \left( 2S + \sum_{p=1}^{S} \frac{\min\{k, S\}}{p} \right) + \tau_k$$

$$\leq 2 \left( \sum_{i=1}^{k} \frac{1}{\tau_i} \right)^{-1} \left( 2S + \sum_{p=1}^{S} \frac{\min\{n, S\}}{p} \right) + \tau_k$$

$$= 2 \left( \sum_{i=1}^{k} \frac{1}{\tau_i} \right)^{-1} \left( 2S + \sum_{p=\min\{n,S\}+1}^{S} \frac{\min\{n, S\}}{p} + \min\{n, S\} \sum_{p=1}^{\min\{n,S\}} \frac{1}{p} \right) + \tau_k$$

$$\leq 2 \left( \sum_{i=1}^{k} \frac{1}{\tau_i} \right)^{-1} \left( 3S + \min\{n, S\} \sum_{p=1}^{\min\{n,S\}} \frac{1}{p} \right) + \tau_k$$

$$\leq 2 \left( \sum_{i=1}^{k} \frac{1}{\tau_i} \right)^{-1} \left( 3S + \min\{n, S\} \left( 2 + \log\left(\min\{n, S\}\right) \right) \right) + \tau_k$$

$$\leq 10 \left( \sum_{i=1}^{k} \frac{1}{\tau_i} \right)^{-1} \left( S + \min\{n, S\} \log\left(\min\{n, S\}\right) \right) + \tau_k,$$

where we use the standard bound on the harmonic series. Thus, the expectation of the total time (15) can be bounded by

$$\mathbb{E}\left[\bar{T}\right] \leq 10 \left( \sum_{i=1}^{k} \frac{1}{\tau_i} \right)^{-1} \left( S + \min\{n, S\} \log\left(\min\{n, S\}\right) \right) + 2\tau_k$$

$$\leq 10 \left( \sum_{i=1}^{k} \frac{1}{\tau_i} \right)^{-1} \left( S + \min\{n, S\} \log\left(\min\{n, S\}\right) + k \right) + 2\tau_k,$$

where in the last line we add $k \geq 0$. Recall that $k$ is a parameter we can choose. Let us take

$$k = \arg\min_{j \in [n]} \left( \sum_{i=1}^{j} \frac{1}{\tau_i} \right)^{-1} \left( S + \min\{n, S\} \log\left(\min\{n, S\}\right) + j \right).$$

Using Lemma 4, we have

$$\tau_k \leq \min_{j \in [n]} \left( \sum_{i=1}^{j} \frac{1}{\tau_i} \right)^{-1} \left( S + \min\{n, S\} \log\left(\min\{n, S\}\right) + j \right)$$

and hence

$$\mathbb{E}\left[\bar{T}\right] \leq 12 \min_{j \in [n]} \left( \left( \sum_{i=1}^{j} \frac{1}{\tau_i} \right)^{-1} \left( S + \min\{n, S\} \log\left(\min\{n, S\}\right) + j \right) \right).$$

$\square$

# D  Proofs for Algorithm 1 (Freya PAGE)

The proofs use the simplified notation $t^*(S) := t^*(S, [\tau_i]_{i=1}^n)$ from Definition 3.

Since the update rule of PAGE coincides with that of Freya PAGE, one can directly apply the iteration complexity results established in Tyurin et al. [2023].

**Theorem 4** (Iteration complexity). *Let Assumptions 1, 2 and 3 hold. Consider any minibatch size $S \in \mathbb{N} := \{1, 2, \ldots\}$, any probability $p \in (0, 1]$, and let the stepsize be $\gamma = \left(L_- + L_\pm \sqrt{\frac{1-p}{pS}}\right)^{-1}$. Then, after*

$$K \geq K_{\mathsf{PAGE}} := \frac{2\delta^0}{\varepsilon}\left(L_- + L_\pm\sqrt{\frac{1-p}{pS}}\right) \tag{7}$$

*iterations of Algorithm 1, we have $\mathbb{E}\left[\left\|\nabla f(\hat{x}^K)\right\|^2\right] \leq \varepsilon$, where $\hat{x}^K$ is sampled uniformly at random from the iterates $\{x^0, \ldots, x^{K-1}\}$.*

*Proof.* The result follows from Theorem 6 of Tyurin et al. [2023], using the parameters from the "Uniform With Replacement" line in Table 1 of the same work. □

**Theorem 5** (Time complexity with free parameters $p$ and $S$). *Consider the assumptions and the parameters from Theorem 4, plus Assumption 4. The expected time complexity of Algorithm 1 is at most*

$$\begin{aligned}T(p, S, [\tau_i]_{i=1}^n) := {} & 12 \cdot t^*(m, [\tau_i]_{i=1}^n) \\ & + \frac{48\delta^0}{\varepsilon}\left(L_- + L_\pm\sqrt{\frac{1-p}{pS}}\right) \times \left\{p \cdot t^*(m, [\tau_i]_{i=1}^n) + (1-p) \cdot t^*(S, [\tau_i]_{i=1}^n)\right\}.\end{aligned} \tag{8}$$

*Proof.* The result established in Theorem 4 says that the iteration complexity of the algorithm is

$$K_{\mathsf{PAGE}} := \frac{2\delta^0}{\varepsilon}\left(L_- + L_\pm\sqrt{\frac{1-p}{pS}}\right).$$

At each iteration, with probability $1 - p$, the workers compute $S$ differences of stochastic gradients, which by Theorem 2 takes

$$4\min_{j \in [n]}\left(\left(\sum_{i=1}^j \frac{1}{\tau_i}\right)^{-1}(S + j)\right)$$

seconds. Otherwise, they collect the full gradient, which can be done (Theorem 1) in

$$12\min_{j \in [n]}\left(\left(\sum_{i=1}^j \frac{1}{\tau_i}\right)^{-1}(m + \min\{m, n\}\log(\min\{m, n\}) + j)\right)$$

$$\leq 24\min_{j \in [n]}\left(\left(\sum_{i=1}^j \frac{1}{\tau_i}\right)^{-1}(m + j)\right)$$

seconds, where the inequality uses Assumption 4. Hence, recalling the notation

$$t^*(S) := \min_{j \in [n]}\left(\left(\sum_{i=1}^j \frac{1}{\tau_i}\right)^{-1}(S + j)\right),$$

the (expected) time complexity of the method is

$$T(p, S, [\tau_i]_{i=1}^n) = 24t^*(m) + 24K_{\mathsf{PAGE}} \times (pt^*(m) + (1-p)t^*(S)),$$

where the term $t^*(m)$ corresponds to the preprocessing step, when the algorithm needs to calculate $g^0 = \nabla f(x^0) = 1/m \sum_{i=1}^m \nabla f_i(x^0)$. □

**Theorem 13.** *Up to a constant factor, the time complexity $T(p, S, [\tau_i]_{i=1}^n)$ from (8) is at least*

$$t^*(m) + \frac{\delta^0}{\varepsilon} \min \left\{ L_- t^*(m), L_- t^*(S) + L_\pm \sqrt{\frac{t^*(m)t^*(S)}{S}} \right\}, \qquad (17)$$

*and attains this value with*

$$p^*(S) = \begin{cases} 1, & L_- t^*(m) \le L_- t^*(S) + L_\pm \sqrt{\frac{t^*(m)t^*(S)}{S}} \\ \frac{t^*(S)}{t^*(m)}, & \text{otherwise.} \end{cases} \qquad (18)$$

*Proof.* Up to a constant factor, by Theorem 5, the time complexity of Freya PAGE is

$$T(p, S, [\tau_i]_{i=1}^n) := t^*(m) + \frac{\delta^0}{\varepsilon} \left( L_- + L_\pm \sqrt{\frac{1-p}{pS}} \right) (pt^*(m) + (1-p)t^*(S)).$$

Let us denote the second term in the above equation as $T_{p,S}$. Then for all $p \ge \frac{1}{2}$, we have

$$T_{p,S} \propto \frac{\delta^0}{\varepsilon} \left( L_- + L_\pm \sqrt{\frac{1-p}{pS}} \right) (pt^*(m) + (1-p)t^*(S)) \ge \frac{\delta^0}{2\varepsilon} L_- t^*(m), \qquad (19)$$

and for all $S \ge \frac{m}{2}$

$$T_{p,S} \ge \frac{\delta^0}{\varepsilon} L_- \left\{ pt^*(m) + (1-p) \min_{j \in [n]} \left( \left( \sum_{i=1}^j \frac{1}{\tau_i} \right)^{-1} \left( \frac{m}{2} + j \right) \right) \right\} \ge \frac{\delta^0}{2\varepsilon} L_- t^*(m).$$

Otherwise, when $p < \frac{1}{2}$ and $S < \frac{m}{2}$, we have

$$\begin{aligned} T_{p,S} &\ge \frac{\delta^0}{2\varepsilon} \left( L_- + L_\pm \sqrt{\frac{1}{pS}} \right) (pt^*(m) + t^*(S)) \\ &\ge \frac{\delta^0}{2\varepsilon} L_- t^*(S) + \frac{\delta^0}{2\varepsilon} \left( L_\pm \sqrt{\frac{1}{pS}} \right) (pt^*(m) + t^*(S)) \\ &\ge \frac{\delta^0}{2\varepsilon} L_- t^*(S) + \frac{\delta^0}{\varepsilon} \left( L_\pm \sqrt{\frac{1}{pS}} \right) \sqrt{pt^*(m)t^*(S)} \\ &= \frac{\delta^0}{2\varepsilon} L_- t^*(S) + \frac{\delta^0}{\varepsilon} L_\pm \sqrt{\frac{t^*(m)t^*(S)}{S}}. \end{aligned}$$

Hence, up to a constant factor,

$$T(p, S, [\tau_i]_{i=1}^n) = t^*(m) + T_{p,S} \ge t^*(m) + \frac{\delta^0}{\varepsilon} \min \left\{ L_- t^*(m), L_- t^*(S) + L_\pm \frac{\sqrt{t^*(m)t^*(S)}}{\sqrt{S}} \right\}.$$

It can be easily verified that this bound can be attained (up to a constant factor) using the parameter $p$ as defined in (18). $\qquad \square$

**Theorem 14.** *Up to a constant factor, the minimum of the time complexities (8) and (17) is*

$$t^*(m) + \frac{\delta^0}{\varepsilon} \min \left\{ L_- t^*(m), \min_{S \in [m]} \left[ L_- t^*(S) + L_\pm \frac{\sqrt{t^*(m)t^*(S)}}{\sqrt{S}} \right] \right\}$$

*and is achieved for*

$$S^* = \arg \min_{S \in [m]} \min \left\{ L_- t^*(m), L_- t^*(S) + L_\pm \frac{\sqrt{t^*(m)t^*(S)}}{\sqrt{S}} \right\}$$

*and $p^* = p^*(S^*)$, where $p^*(S)$ is defined in (18).*

*Proof.* This is a simple corollary of Theorem 13: we take $S$ that minimizes (17). $\qquad \square$

In certain scenarios, we can derive the optimal parameter values explicitly.

**Theorem 15.**

1. *If $n \leq \frac{L_{\pm}\sqrt{m}}{L_-} \leq m$, then (up to constants) $S^* = \frac{L_{\pm}\sqrt{m}}{L_-}$ and $p^* = \frac{L_{\pm}}{L_-\sqrt{m}}$ are optimal parameters and*

$$T(p^*, S^*, [\tau_i]_{i=1}^n) = t^*(m) + \frac{\delta^0 L_-}{\varepsilon} t^*\left(\frac{L_{\pm}\sqrt{m}}{L_-}\right).$$

2. *If $\frac{L_{\pm}\sqrt{m}}{L_-} \leq 1$, then (up to constants) $S^* = 1$ and $p^* = 1/m$ are optimal parameters and*

$$T(p^*, S^*, [\tau_i]_{i=1}^n) = t^*(m) + \frac{\delta^0 L_-}{\varepsilon} t^*(1).$$

3. *If $\frac{L_{\pm}\sqrt{m}}{L_-} \geq m$, then (up to constants) $p^* = 1$ is an optimal parameter and*

$$T(p^*, S, [\tau_i]_{i=1}^n) = \frac{\delta^0 L_-}{\varepsilon} t^*(m)$$

*for any $S \in [m]$.*

*Proof.* We fist consider the case when $n \leq \frac{L_{\pm}\sqrt{m}}{L_-} \leq m$. Since $j \leq n \leq \frac{L_{\pm}\sqrt{m}}{L_-}$, we have

$$\frac{L_{\pm}\sqrt{m}}{L_-} + \frac{L_{\pm}}{L_-\sqrt{m}} j \geq \frac{1}{2}\left(\frac{L_{\pm}\sqrt{m}}{L_-} + j\right), \tag{20}$$

and from the assumption that $L_- \geq \frac{L_{\pm}}{\sqrt{m}}$ it follows that

$$\frac{L_{\pm}\sqrt{m}}{L_-} + j \geq \frac{L_{\pm}\sqrt{m}}{L_-} + \frac{L_{\pm}}{L_-\sqrt{m}} j \tag{21}$$

for all $j \in [n]$. Thus,

$$\frac{L_{\pm}}{L_-\sqrt{m}} t^*(m) = \min_{j \in [n]} \left(\left(\sum_{i=1}^j \frac{1}{\tau_i}\right)^{-1}\left(\frac{L_{\pm}\sqrt{m}}{L_-} + \frac{L_{\pm}}{L_-\sqrt{m}} j\right)\right) \overset{(20)}{\geq} \frac{1}{2} t^*\left(\frac{L_{\pm}\sqrt{m}}{L_-}\right),$$

and

$$\frac{L_{\pm}}{L_-\sqrt{m}} t^*(m) = \min_{j \in [n]} \left(\left(\sum_{i=1}^j \frac{1}{\tau_i}\right)^{-1}\left(\frac{L_{\pm}\sqrt{m}}{L_-} + \frac{L_{\pm}}{L_-\sqrt{m}} j\right)\right) \overset{(21)}{\leq} t^*\left(\frac{L_{\pm}\sqrt{m}}{L_-}\right).$$

It follows that

$$\frac{1}{2} t^*\left(\frac{L_{\pm}\sqrt{m}}{L_-}\right) \leq \frac{L_{\pm}}{L_-\sqrt{m}} t^*(m) \leq t^*\left(\frac{L_{\pm}\sqrt{m}}{L_-}\right). \tag{22}$$

Since $\frac{L_{\pm}\sqrt{m}}{L_-} \leq m$, we have

$$\min_{S \in [m]}\left\{L_- t^*(S) + L_{\pm}\frac{\sqrt{t^*(m)t^*(S)}}{\sqrt{S}}\right\} \leq L_- \min_{S \in [m]}\left\{t^*(S) + \sqrt{m}\frac{\sqrt{t^*(m)t^*(S)}}{\sqrt{S}}\right\}$$

$$\leq 2L_- t^*(m), \tag{23}$$

and thus, according to the result from Theorem 14, it is sufficient to minimize

$$t'(S) := L_- t^*(S) + L_{\pm}\frac{\sqrt{t^*(m)t^*(S)}}{\sqrt{S}}.$$

First, let us note that

$$t'\left(\frac{L_\pm\sqrt{m}}{L_-}\right) = L_-t^*\left(\frac{L_\pm\sqrt{m}}{L_-}\right) + L_\pm\sqrt{\frac{L_-}{L_\pm\sqrt{m}}t^*(m)t^*\left(\frac{L_\pm\sqrt{m}}{L_-}\right)}$$

$$\overset{(22)}{\leq} L_-t^*\left(\frac{L_\pm\sqrt{m}}{L_-}\right) + L_\pm\sqrt{\frac{L_-}{L_\pm\sqrt{m}}\frac{L_-\sqrt{m}}{L_\pm}t^*\left(\frac{L_\pm\sqrt{m}}{L_-}\right)t^*\left(\frac{L_\pm\sqrt{m}}{L_-}\right)}$$

$$= 2L_-t^*\left(\frac{L_\pm\sqrt{m}}{L_-}\right).$$

If $S \geq \frac{L_\pm\sqrt{m}}{L_-}$, then

$$t'(S) = L_-t^*(S) + L_\pm\sqrt{\frac{t^*(m)t^*(S)}{S}} \geq L_-t^*\left(\frac{L_\pm\sqrt{m}}{L_-}\right).$$

Otherwise, if $S < \frac{L_\pm\sqrt{m}}{L_-}$, then

$$t'(S) = L_-t^*(S) + L_\pm\sqrt{t^*(m)}\sqrt{\min_{j\in[n]}\left(\left(\sum_{i=1}^{j}\frac{1}{\tau_i}\right)^{-1}\left(1+\frac{j}{S}\right)\right)}$$

$$\geq L_\pm\sqrt{t^*(m)}\sqrt{\min_{j\in[n]}\left(\left(\sum_{i=1}^{j}\frac{1}{\tau_i}\right)^{-1}\left(1+\frac{L_-j}{L_\pm\sqrt{m}}\right)\right)}$$

$$= L_\pm\sqrt{t^*(m)}\sqrt{\frac{L_-}{L_\pm\sqrt{m}}\min_{j\in[n]}\left(\left(\sum_{i=1}^{j}\frac{1}{\tau_i}\right)^{-1}\left(\frac{L_\pm\sqrt{m}}{L_-}+j\right)\right)}$$

$$= L_\pm\sqrt{t^*(m)\frac{L_-}{L_\pm\sqrt{m}}t^*\left(\frac{L_\pm\sqrt{m}}{L_-}\right)}$$

$$\overset{(22)}{\geq} L_\pm\sqrt{\frac{1}{2}t^*\left(\frac{L_\pm\sqrt{m}}{L_-}\right)\frac{L_-\sqrt{m}}{L_\pm}\frac{L_-}{L_\pm\sqrt{m}}t^*\left(\frac{L_\pm\sqrt{m}}{L_-}\right)}$$

$$= \frac{L_-}{\sqrt{2}}t^*\left(\frac{L_\pm\sqrt{m}}{L_-}\right).$$

Therefore, the optimal choice is $S^* = \frac{L_\pm\sqrt{m}}{L_-}$, and by Theorem 13 and inequality (23), $p$ should chosen to be

$$\frac{t^*\left(\frac{L_\pm\sqrt{m}}{L_-}\right)}{t^*(m)}.$$

Using (22), we can conclude that $p^* = \frac{L_\pm}{L_-\sqrt{m}}$ is optimal, proving the first part of the Theorem.

Next, consider the case when $\frac{L_\pm\sqrt{m}}{L_-} \leq 1$. By the reasoning above, it is sufficient to minimize

$$t'(S) := L_-t^*(S) + L_\pm\frac{\sqrt{t^*(m)t^*(S)}}{\sqrt{S}}.$$

First, let us note that

$$t'(1) = L_-t^*(1) + L_\pm\sqrt{t^*(m)t^*(1)} \leq L_-t^*(1) + L_-\sqrt{\frac{t^*(m)t^*(1)}{m}} \leq 2L_-t^*(1),$$

where the last inequality follows from the fact that for any $S \in [m]$, $t^*(S)/s \leq t^*(1)$. On the other hand, if $S \geq 1$, then

$$t'(S) = L_-t^*(S) + L_\pm\frac{\sqrt{t^*(m)t^*(S)}}{\sqrt{S}} \geq L_-t^*(S) \geq L_-t^*(1).$$

Therefore, the optimal choice is $S^* = 1$. Then

$$\frac{\delta^0}{\varepsilon} \left( L_- + L_\pm \sqrt{m-1} \right) \left( \frac{t^*(m)}{m} + \left( 1 - \frac{1}{m} \right) t^*(1) \right)$$

$$\leq \frac{2\delta^0}{\varepsilon} \left( L_- + \frac{L_-}{\sqrt{m}} \sqrt{m-1} \right) t^*(1)$$

$$\leq \frac{4\delta^0 L_-}{\varepsilon} t^*(1),$$

while for any $p$

$$\frac{\delta^0}{\varepsilon} \left( L_- + L_\pm \sqrt{\frac{1-p}{p}} \right) (pt^*(m) + (1-p)t^*(1))$$

$$\geq \frac{\delta^0 L_-}{\varepsilon} (pt^*(m) + (1-p)t^*(1))$$

$$\geq \frac{\delta^0 L_-}{\varepsilon} (pt^*(1) + (1-p)t^*(1))$$

$$= \frac{\delta^0 L_-}{\varepsilon} t^*(1).$$

Hence $p^* = 1/m$.

It remains to prove the third result. Suppose that $L_- < \frac{L_\pm}{\sqrt{m}}$. Then, the last part of the theorem follows from the fact that

$$\min_{S \in [m]} \left\{ L_- t^*(S) + L_\pm \frac{\sqrt{t^*(m)t^*(S)}}{\sqrt{S}} \right\} \geq L_\pm \min_{S \in [m]} \left\{ \frac{\sqrt{t^*(m)t^*(S)}}{\sqrt{S}} \right\}$$

$$= L_\pm \left\{ \frac{\sqrt{t^*(m)t^*(m)}}{\sqrt{m}} \right\}$$

$$\geq L_- t^*(m).$$

$\square$

In practice, the values of smoothness constants are often unknown. However, the algorithm can still be run with close to optimal parameters.

**Theorem 7** (Main result in the large-scale regime). *Consider the assumptions and parameters from Theorem 4, plus Assumption 4. Up to a constant factor and smoothness constants, if $\sqrt{m} \geq n$, then the optimal choice of parameters in (8) is $S^* = \lceil \sqrt{m} \rceil$ and $p^* = 1/\sqrt{m}$. For this choice, the expected time complexity of Algorithm 1 is at most*

$$T(1/\sqrt{m}, \sqrt{m}, [\tau_i]_{i=1}^n) = 12t^*(m, [\tau_i]_{i=1}^n) + \frac{192\delta^0 \max\{L_-, L_\pm\}}{\varepsilon} t^*(\sqrt{m}, [\tau_i]_{i=1}^n) \qquad (10)$$

*seconds. The iteration complexity with $S = \lceil \sqrt{m} \rceil$ and $p = 1/\sqrt{m}$ is $K_{\mathsf{PAGE}} \leq 4\delta^0 \max\{L_-, L_\pm\}/\varepsilon$.*

*Proof.* The proof is the same as in Theorem 15. Indeed, up to a constant factor, the time complexity (8) can be bounded as

$$T(p, S, [\tau_i]_{i=1}^n) = t^*(m) + \frac{\delta^0}{\varepsilon} \left( L_- + L_\pm \sqrt{\frac{1-p}{pS}} \right) (pt^*(m) + (1-p)t^*(S))$$

$$\leq t^*(m) + \frac{2\delta^0 \max\{L_-, L_\pm\}}{\varepsilon} \left( 1 + \sqrt{\frac{1-p}{pS}} \right) (pt^*(m) + (1-p)t^*(S)) .$$

Therefore, by setting $L_\pm = L_-$ in Theorem 15, one can easily derive the parameters $p$ and $S$ that are optimal up to the smoothness constants. The time complexity (10) can be obtained by applying $S = \lceil \sqrt{m} \rceil$ and $p = 1/\sqrt{m}$ to (8). $\square$

# E Freya PAGE with Other Samplings

Algorithm 1 can be adapted to accommodate other sampling methods. This brings us to the introduction of Algorithm 6, which supports virtually any sampling strategy, formalized by the following mapping:

**Definition 16** (Sampling). A *sampling* is a random mapping $\mathbf{S}_S$, which takes as an input a set of indices $\mathcal{I} := \{a_1, \ldots, a_m\}$ and returns a (multi)set $\{a_{i_1}, \ldots, a_{i_S}\}$, where $a_{i_j} \in \mathcal{I}$ for all $j \in [S]$.

---

**Algorithm 6** Freya PAGE (with virtually any sampling)

---

1: **Parameters:** starting point $x^0 \in \mathbb{R}^d$, learning rate $\gamma > 0$, minibatch size $S$, sampling $\mathbf{S}_S$, probability $p \in (0, 1]$, initialization $g^0 = \nabla f(x^0)$ using ComputeGradient$([m], x^0)$   (Alg. 2)
2: **for** $k = 0, 1, \ldots, K - 1$ **do**
3:     $x^{k+1} = x^k - \gamma g^k$
4:     Sample $c^k \sim \text{Bernoulli}(p)$
5:     **if** $c^k = 1$ **then**
6:         $\nabla f(x^{k+1}) = \text{ComputeGradient}(x^{k+1})$   (Alg. 2)
7:         $g^{k+1} = \nabla f(x^{k+1})$
8:     **else**
9:         Sample indices $\mathcal{S}^k = \mathbf{S}_S([m])$
10:        $\frac{1}{S} \sum_{i \in \mathcal{S}^k} \left( \nabla f_i(x^{k+1}) - \nabla f_i(x^k) \right)$
          $= \text{ComputeBatchDifferenceAnySampling}(\mathcal{S}^k, x^{k+1}, x^k)$   (Alg. 7)
11:        $g^{k+1} = g^k + \frac{1}{S} \sum_{i \in \mathcal{S}^k} \left( \nabla f_i(x^{k+1}) - \nabla f_i(x^k) \right)$
12:     **end if**
13: **end for**

---

**Algorithm 7** ComputeBatchDifferenceAnySampling$(\mathcal{S}, x, y)$

---

1: **Input:** multiset $\mathcal{S}$, points $x, y \in \mathbb{R}^d$
2: Init $g = 0 \in \mathbb{R}^d$, multiset $\mathcal{M} = \emptyset$
3: Broadcast $x$ to all workers
4: For each worker, sample $j$ from $\mathcal{S}$ (uniformly) and ask it to calculate $\nabla f_j(x) - \nabla f_j(y)$
5: **while** $\mathcal{M} \neq \mathcal{S}$ **do**
6:     Wait for $\nabla f_p(x) - \nabla f_p(y)$ from a worker
7:     **if** $p \in \mathcal{S} \backslash \mathcal{M}$ **then**
8:         $g \leftarrow g + \frac{1}{|\mathcal{S}|} \left( \nabla f_p(x) - \nabla f_p(y) \right)$
9:         Update $\mathcal{M} \leftarrow \mathcal{M} \cup \{p\}$
10:     **end if**
11:     Sample $j$ from $\mathcal{S} \backslash \mathcal{M}$ (uniformly) and ask this worker to calculate $\nabla f_j(x) - \nabla f_j(y)$
12: **end while**
13: Return $g = \frac{1}{|\mathcal{S}|} \sum_{i \in \mathcal{S}} \left( \nabla f_i(x) - \nabla f_i(y) \right)$

---

The only difference is that instead of ComputeBatchDifference (Algorithm 5), Algorithm 6 uses a new subroutine, called ComputeBatchDifferenceAnySampling (Algorithm 3).

For this algorithm, we can prove the following time complexity guarantees.

**Theorem 17.** *The expected time needed by Algorithm 7 to calculate* $g = \frac{1}{|\mathcal{S}|} \sum_{i \in \mathcal{S}} (\nabla f_i(x) - \nabla f_i(y))$

*is at most*

$$24 \min_{j \in [n]} \left( \left( \sum_{i=1}^{j} \frac{1}{\tau_i} \right)^{-1} (|\mathcal{S}| + \min\{|\mathcal{S}|, n\} \log (\min\{|\mathcal{S}|, n\}) + j) \right) \tag{24}$$

*seconds.*

*Proof.* The proof of this theorem is the same as the proof of Theorem 12. We only have to multiply (14) by 2 because Algorithm 3 calculates $\nabla f_i(x) - \nabla f_i(y)$ instead of $\nabla f_i(x)$. □

While changing the sampling strategy might affect the *iteration complexity* of the method, for a fixed minibatch size $S$, the *time complexity of a single iteration* remains unchanged. Thus, having established the expected time needed by the algorithm to perform a single iteration (i.e., to collect a minibatch of stochastic gradients of the required size), one can simply multiply it by the iteration complexity of the method determined for any supported sampling technique to obtain the resulting time complexity.

With this in mind, we now analyse the time complexity of Algorithm 6 with 2 different sampling techniques: nice sampling and importance sampling. However, it can be analyzed with virtually any other unbiased sampling [Tyurin et al., 2023].

### E.1 Nice sampling

*Nice sampling* returns a random subset of fixed cardinality $S$ chosen uniformly from $[m]$. Unlike *uniform sampling with replacement* used in Algorithm 5, which returns a random multiset (that can include repetitions), the samples obtained by nice sampling are distinct. The iteration complexity of Algorithm 6 with nice sampling is given by the following theorem.

**Theorem 18** (Tyurin et al. [2023], Section 3). *Let Assumptions 1, 2 and 3 hold. Choose a minibatch size $S \in [m]$, a probability $p \in (0, 1]$ and the stepsize*

$$\gamma = \frac{1}{L_- + L_\pm \sqrt{\frac{(1-p)(m-S)}{p(m-1)S}}}.$$

*Then, the number of iteration needed by Algorithm 6 with* nice sampling *to reach an $\varepsilon$-stationary point is*

$$K = \frac{2\delta^0}{\varepsilon} \left( L_- + L_\pm \sqrt{\frac{(1-p)(m-S)}{p(m-1)S}} \right). \tag{25}$$

*Proof.* The result follows from Theorem 6 and Table 1 from [Tyurin et al., 2023]. □

**Theorem 19.** *Consider the assumptions and parameters from Theorem 18 and Assumption 4. Up to a constant factor, the time complexity of Algorithm 6 is*

$$T(p, S, [\tau_i]_{i=1}^n) = t^*(m, [\tau_i]_{i=1}^n) + \frac{\delta^0}{\varepsilon} \left( L_- + L_\pm \sqrt{\frac{(1-p)(m-S)}{p(m-1)S}} \right) \times$$

$$\times \left\{ p \times t^*(m, [\tau_i]_{i=1}^n) + (1-p) \times t^*(S + \min\{S, n\} \log(\min\{S, n\}), [\tau_i]_{i=1}^n) \right\}, \tag{26}$$

*where $t^*$ is defined from Definition 3.*

**Remark 20.** *Compared to Theorem 5, which uses* uniform sampling with replacement*, the guarantees for* nice sampling *are slightly worse: the term $t^*(S, [\tau_i]_{i=1}^n)$ from Theorem 5 here is replaced with $t^*(S + \min\{S, n\} \log(\min\{S, n\}), [\tau_i]_{i=1}^n)$. Ignoring the logarithmic term $\log(\min\{S, n\})$ ($\leq \log(\min\{m, n\})$), the result from Theorem 19 is equivalent to that in Theorem 5. Thus, Theorems 6 and 7 hold also for the nice sampling (up to logarithmic factors).*

*Proof.* We use the same reasoning as in the proof of Theorem 5. With probability $p$, the algorithm calculates the full gradients, which by Theorem 1 requires

$$\Theta \left( \min_{j \in [n]} \left( \left( \sum_{i=1}^j \frac{1}{\tau_i} \right)^{-1} (m + \min\{m, n\} \log(\min\{m, n\}) + j) \right) \right)$$

$$= \Theta \left( \min_{j \in [n]} \left( \left( \sum_{i=1}^j \frac{1}{\tau_i} \right)^{-1} (m + j) \right) \right)$$

seconds, where we use Assumption 4. With probability $1 - p$, the algorithm calls ComputeBatchDif-ferenceAnySampling, which by Theorem 17 requires

$$\Theta\left(\min_{j\in[n]}\left(\left(\sum_{i=1}^{j}\frac{1}{\tau_i}\right)^{-1}(S + \min\{S, n\}\log(\min\{S, n\}) + j)\right)\right)$$

seconds. One can obtain the result by multiplying the iteration complexity (25) by the expected time needed to collect the required number of stochastic gradients per iteration and adding the preprocessing time. □

## E.2 Importance sampling

Here we additionally assume $L_i$-smoothness of the local objective functions $f_i$.

**Assumption 5.** *The functions $f_i$ are $L_i$-smooth. We denote $\bar{L} := \frac{1}{m}\sum_{j=1}^{m} L_j$ and $L_{\max} := \max_{i\in[n]} L_i$.*

Importance sampling is a sampling technique that returns a multiset of indices *with repetitions*. Index $j$ is included in the multiset with probability $\frac{L_j}{\sum_{i=1}^{n} L_i}$.

**Theorem 21** (Tyurin et al. [2023], Section 3). *Let Assumptions 1 and 5 hold. Choose a minibatch size $S \in [m]$, probability $p \in (0, 1]$ and the stepsize*

$$\gamma = \frac{1}{L_- + \bar{L}\sqrt{\frac{1-p}{pS}}}.$$

*Then, the number of iteration needed by Algorithm 1 with importance sampling to reach an $\varepsilon$-stationary point is*

$$K = \frac{2\delta^0}{\varepsilon}\left(L_- + \bar{L}\sqrt{\frac{1-p}{pS}}\right). \tag{27}$$

The complexity (27) is nearly identical to (7) and (25), with the only difference being the dependence on $\bar{L}$ rather than $L_\pm$. Thus, all the results up to constant and logarithmic factors can be derived using the same methodology as that outlined in Section 4, with the simple substitution of $L_\pm$ with $\bar{L}$.

## F Dynamic Bounds

As noted in Section 4.4, the results from Section D can be easily generalized to iteration-dependent processing times.

**Theorem 22.** *Consider the assumptions and the parameters from Theorem 4 and Assumption 4. Up to a constant factor, the time complexity of* Freya PAGE *(Algorithm 1) with iteration-dependent processing times $\{\tau_i^k\}$, which are defined in Section 4.4, is at most*

$$\min_{j\in[n]}\left(\left(\sum_{i=1}^{j}\frac{1}{\tau_{\pi_{-1,i}}^{-1}}\right)^{-1}(m + j)\right)$$

$$+ \sum_{k=0}^{\lceil K_{\mathsf{PAGE}}\rceil}\left\{p\min_{j\in[n]}\left(\left(\sum_{i=1}^{j}\frac{1}{\tau_{\pi_{k,i}}^{k}}\right)^{-1}(m + j)\right) + (1-p)\min_{j\in[n]}\left(\left(\sum_{i=1}^{j}\frac{1}{\tau_{\pi_{k,i}}^{k}}\right)^{-1}(S + j)\right)\right\}, \tag{28}$$

*to find an $\varepsilon$-stationary point, where $p \in (0, 1]$ and $S \in \mathbb{N}$ are free parameters, and $\pi_{k,\cdot}$ is a permutation such that $\tau_{\pi_{k,1}}^{k} \leq \cdots \leq \tau_{\pi_{k,n}}^{k}$ for all $k \geq -1$.*

**Remark 23.** *The theorem can be trivially extended to other samplings by changing $K_{\mathsf{PAGE}}$ to the iteration complexities from Theorems 18 and 21.*

*Proof.* The reasoning behind this result is exactly the same as in the proof of Theorem 5. The only difference is that in this more general setting, the expected time per iteration varies across iterations. Therefore, instead of simply multiplying, one needs to sum over the iterations to obtain the total time complexity.

We introduce the permutations to ensure that $\{\tau^k_{\pi_{k,i}}\}^n_{i=1}$ are sorted. When $\tau^k_i = \tau_i$, there is no need to introduce them because, throughout the paper, it is assumed $\tau_1 \leq \ldots \leq \tau_n$ (see Section 1). $\square$

# G Examples

Here we provide the proofs for the examples from Section 4.3. We will use the notation

$$t(S, j) := \left( \sum_{i=1}^{j} \frac{1}{\tau_i} \right)^{-1} (S + j)$$

for a fixed $S \in [m]$ and all $j \in [n]$.

**Example 1.** *[Equally Fast Workers] Suppose that the upper bounds on the processing times are equal, i.e., $\tau_j = \tau$ for all $j \in [n]$. Then*

$$T(1/\sqrt{m}, \sqrt{m}, [\tau_i]_{i=1}^{n}) = \Theta \left( \tau \max \left\{ \frac{m}{n}, 1 \right\} + \tau \frac{\delta^0 \max\{L_-, L_\pm\}}{\varepsilon} \max \left\{ \frac{\sqrt{m}}{n}, 1 \right\} \right).$$

*Proof.* First, when $\tau_j = \tau$ for all $j \in [n]$, then for any $S \in [m]$, $t(S, j)$ is minimized by taking $j = n$:

$$t^*(S) := \min_{j \in [n]} t(S, j) = \min_{j \in [n]} \left( \left( \sum_{i=1}^{j} \frac{1}{\tau_i} \right)^{-1} (S + j) \right)$$

$$= \min_{j \in [n]} \left( \frac{\tau}{j} (S + j) \right) = \Theta \left( \tau \max \left\{ \frac{S}{n}, 1 \right\} \right).$$

It remains to substitute this equality in (10).

$\square$

**Example 2.** *[Infinitely Fast Worker] If $\tau_1 = 0$, then $T(1/\sqrt{m}, \sqrt{m}, [\tau_i]_{i=1}^{n}) = 0$.*

*Proof.* The statement follows easily from the fact that for any $S \in [m]$ we have

$$t^*(S) := \min_{j \in [n]} \left( \left( \sum_{i=1}^{j} \frac{1}{\tau_i} \right)^{-1} (S + j) \right) \leq \left( \frac{1}{\tau_1} \right)^{-1} (S + j) = 0.$$

$\square$

**Example 3.** *[Infinitely Slow Workers] If $\tau_j = \infty \; \forall j \in [n]$, then $T(1/\sqrt{m}, \sqrt{m}, [\tau_i]_{i=1}^{n}) = \infty$.*

*Proof.* This follows from the fact that for any $S \in [m]$ and any $j \in [n]$ we have

$$t(S, j) := \left( \sum_{i=1}^{j} \frac{1}{\tau_i} \right)^{-1} (S + j) = \infty.$$

$\square$

**Example 4.** *[Extremely Slow Workers] Suppose that the times $\tau_j < \infty$ are fixed $\forall j \leq j_B$ and $\tau_j \geq B \; \forall j > j_B$ for some $B$ large enough. Then $T(1/\sqrt{m}, \sqrt{m}, [\tau_i]_{i=1}^{n}) = T(1/\sqrt{m}, \sqrt{m}, [\tau_i]_{i=1}^{j_B})$.*

*Proof.* Suppose that $B \geq \frac{m + j_B}{\sum_{i=1}^{j_B} \frac{1}{\tau_i}}$ and fix any $k > j_B$. Then, since $\tau_j \geq B$ for all $j > j_B$, we have

$$\frac{k - j_B}{\sum_{i=j_B+1}^{k} \frac{1}{\tau_i}} \geq \frac{k - j_B}{\sum_{i=j_B+1}^{k} \frac{1}{B}} = B \geq \frac{m + j_B}{\sum_{i=1}^{j_B} \frac{1}{\tau_i}} \geq \frac{S + j_B}{\sum_{i=1}^{j_B} \frac{1}{\tau_i}} = t(S, j_B)$$

for any $S \in [m]$. Rearranging and adding $(S + j_B) \sum_{i=1}^{j_B} \frac{1}{\tau_i}$ to both sides of the inequality, we obtain

$$(S + j_B) \left( \sum_{i=1}^{j_B} \frac{1}{\tau_i} \right) + (S + j_B) \left( \sum_{i=j_B+1}^{k} \frac{1}{\tau_i} \right) \leq (S + j_B) \left( \sum_{i=1}^{j_B} \frac{1}{\tau_i} \right) + (k - j_B) \left( \sum_{i=1}^{j_B} \frac{1}{\tau_i} \right),$$

meaning that

$$t(S, j_B) = \frac{S + j_B}{\sum_{i=1}^{j_B} \frac{1}{\tau_i}} \leq \frac{S + k}{\sum_{i=1}^{k} \frac{1}{\tau_i}} = t(S, k)$$

for any $k > j_B$ and any $S \in [m]$. Therefore,

$$\min_{j \in [n]} \left( \left( \sum_{i=1}^{j} \frac{1}{\tau_i} \right)^{-1} (S + j) \right) = \min_{j \in [j_B]} \left( \left( \sum_{i=1}^{j} \frac{1}{\tau_i} \right)^{-1} (S + j) \right)$$

for any $S \in [m]$, which proves the claim. $\qquad\square$

# H A New Stochastic Gradient Method: Freya SGD

In this section, we introduce a new a non-variance reduced SGD method that we call Freya SGD. Freya SGD is closely aligned with Rennala SGD [Tyurin and Richtárik, 2023], but Freya SGD does not require the $\sigma^2$–bounded variance assumption on stochastic gradients.

---
**Algorithm 8** Freya SGD
---
1: **Parameters:** starting point $x^0 \in \mathbb{R}^d$, learning rate $\gamma > 0$, minibatch size $S$
2: **for** $k = 0, 1, \ldots, K - 1$ **do**
3: $\quad \frac{1}{S} \sum_{i \in \mathcal{S}^k} \nabla f_i(x^k) = \mathsf{ComputeBatch}(S, x^k)$ $\qquad\qquad\qquad\qquad$ (Alg. 4)
4: $\quad x^{k+1} = x^k - \gamma \frac{1}{S} \sum_{i \in \mathcal{S}^k} \nabla f_i(x^k)$
5: **end for**
$\quad$ (note): $\mathcal{S}^k$ is a set of size $\left|\mathcal{S}^k\right| = S$ of i.i.d. indices sampled from $[m]$ *uniformly with replacement*

---

**Assumption 6.** *For all $i \in [n]$, there exists $f_i^*$ such that $f_i(x) \geq f_i^*$ for all $x \in \mathbb{R}^d$.*

We define

$$\Delta^* := \frac{1}{n} \sum_{i=1}^{n} \left(f^* - f_i^*\right).$$

**Theorem 24.** *Let Assumptions 1, 5 and 6 hold. Choose minibatch size $S \in [m]$ and stepsize*

$$\gamma = \min \left\{ \frac{\sqrt{S}}{\sqrt{LL_{\max}K_{\mathrm{SGD}}}}, \frac{1}{L\left(1 - \frac{1}{S}\right)}, \frac{S\varepsilon}{4LL_{\max}\Delta^*} \right\},$$

*where*

$$K_{\mathrm{SGD}} := \frac{12\delta^0 L}{\varepsilon} \max \left\{ 1 - \frac{1}{S}, \frac{12L_{\max}\delta^0}{S\varepsilon}, \frac{4L_{\max}\Delta^*}{S\varepsilon} \right\}.$$

*Then, the number of iterations needed by Algorithm 8 to reach an $\varepsilon$-stationary point is $\lceil K_{\mathrm{SGD}} \rceil$.*

*Proof.* The iteration complexity can be proved using Corollary 1 and Proposition 3 (i) of Khaled and Richtárik [2022] (with $q_i = 1/n$). $\qquad\qquad\qquad\qquad\qquad\qquad\qquad\qquad\qquad\qquad\square$

**Theorem 25.** *Consider the assumptions and the parameters from Theorem 24. Up to a constant factor, the time complexity of Freya SGD (Algorithm 8) is at most*

$$T_{\mathrm{SGD}}(S, [\tau_i]_{i=1}^n) := \frac{\delta^0 L_-}{\varepsilon} \left(1 - \frac{1}{S} + \frac{L_{\max}}{\varepsilon S}\left(\delta^0 + \Delta^*\right)\right) \times \min_{j \in [n]} \left( \left(\sum_{i=1}^{j} \frac{1}{\tau_i}\right)^{-1}(S + j) \right)$$

*and is minimized by choosing*

$$S^* = \frac{L_{\max}}{\varepsilon}\left(\delta^0 + \Delta^*\right).$$

*Up to a constant factor, we get*

$$T_{\mathrm{SGD}}(S^*, [\tau_i]_{i=1}^n) = \frac{\delta^0 L_-}{\varepsilon} \min_{j \in [n]} \left( \left(\sum_{i=1}^{j} \frac{1}{\tau_i}\right)^{-1}\left(\frac{L_{\max}}{\varepsilon}\left(\delta^0 + \Delta^*\right) + j\right) \right).$$

*Proof.* At each iteration, the algorithm needs to collect a minibatch of stochastic gradients of size $S$. Multiplying the iteration complexity of Theorem 24 by the time needed to gather such a minibatch (Theorem 11), the resulting time complexity is

$$T_{\mathrm{SGD}}(S, [\tau_i]_{i=1}^n) = \frac{\delta^0 L_-}{\varepsilon} \left(1 - \frac{1}{S} + \frac{L_{\max}}{\varepsilon S}\left(\delta^0 + \Delta^*\right)\right) \times \min_{j \in [n]} \left( \left(\sum_{i=1}^{j} \frac{1}{\tau_i}\right)^{-1}(S + j) \right).$$

We now find the optimal $S$. Assume first that $\frac{L_{\max}}{\varepsilon}\left(\delta^0 + \Delta^*\right) \leq \frac{1}{2}$. Assumption 5 ensures that the function $f$ is $L_{\max}$-smooth. Thus, we have $\left\|\nabla f(x^0)\right\|^2 \leq 2L_{\max}\delta^0 \leq \varepsilon$, and $x^0$ is an $\varepsilon$-solution. Therefore, we can take any $S \geq 1$.

Now, suppose that $\frac{L_{\max}}{\varepsilon}\left(\delta^0 + \Delta^*\right) > \frac{1}{2}$. Then we have

$$T_{\text{SGD}}\left(L_{\max}/\varepsilon\left(\delta^0 + \Delta^*\right), [\tau_i]_{i=1}^n\right) = \frac{\delta^0 L_-}{\varepsilon}\left(2 - \frac{1}{S}\right)\min_{j\in[n]}\left(\left(\sum_{i=1}^j \frac{1}{\tau_i}\right)^{-1}\left(\frac{L_{\max}}{\varepsilon}\left(\delta^0 + \Delta^*\right) + j\right)\right)$$

$$\leq \frac{2\delta^0 L_-}{\varepsilon}\min_{j\in[n]}\left(\left(\sum_{i=1}^j \frac{1}{\tau_i}\right)^{-1}\left(\frac{L_{\max}}{\varepsilon}\left(\delta^0 + \Delta^*\right) + j\right)\right).$$

For all $S > \max\left\{\frac{L_{\max}}{\varepsilon}\left(\delta^0 + \Delta^*\right), 1\right\}$, we get $S \geq 2$ and hence

$$T_{\text{SGD}}(S, [\tau_i]_{i=1}^n) = \frac{\delta^0 L_-}{\varepsilon}\left(1 - \frac{1}{S} + \frac{L_{\max}}{\varepsilon S}\left(\delta^0 + \Delta^*\right)\right)\min_{j\in[n]}\left(\left(\sum_{i=1}^j \frac{1}{\tau_i}\right)^{-1}(S + j)\right)$$

$$\geq \frac{\delta^0 L_-}{2\varepsilon}\min_{j\in[n]}\left(\left(\sum_{i=1}^j \frac{1}{\tau_i}\right)^{-1}(S + j)\right)$$

$$\geq \frac{\delta^0 L_-}{2\varepsilon}\min_{j\in[n]}\left(\left(\sum_{i=1}^j \frac{1}{\tau_i}\right)^{-1}\left(\frac{L_{\max}}{\varepsilon}\left(\delta^0 + \Delta^*\right) + j\right)\right).$$

Let us now consider the case $1 < S \leq \max\left\{\frac{L_{\max}}{\varepsilon}\left(\delta^0 + \Delta^*\right), 1\right\}$. We can additionally assume that $\frac{L_{\max}}{\varepsilon}\left(\delta^0 + \Delta^*\right) > 1$ and $S \leq \frac{L_{\max}}{\varepsilon}\left(\delta^0 + \Delta^*\right)$ (otherwise, the set $S$ that satisfies the condition $1 < S \leq \max\left\{\frac{L_{\max}}{\varepsilon}\left(\delta^0 + \Delta^*\right), 1\right\}$ is empty). We get

$$T_{\text{SGD}}(S, [\tau_i]_{i=1}^n) = \frac{\delta^0 L_-}{\varepsilon}\left(1 - \frac{1}{S} + \frac{L_{\max}}{\varepsilon S}\left(\delta^0 + \Delta^*\right)\right)\min_{j\in[n]}\left(\left(\sum_{i=1}^j \frac{1}{\tau_i}\right)^{-1}(S + j)\right)$$

$$\geq \frac{\delta^0 L_-}{\varepsilon}\left(\frac{L_{\max}}{\varepsilon S}\left(\delta^0 + \Delta^*\right)\right)\min_{j\in[n]}\left(\left(\sum_{i=1}^j \frac{1}{\tau_i}\right)^{-1}(S + j)\right)$$

$$= \frac{\delta^0 L_-}{\varepsilon}\min_{j\in[n]}\left(\left(\sum_{i=1}^j \frac{1}{\tau_i}\right)^{-1}\left(\frac{L_{\max}}{\varepsilon}\left(\delta^0 + \Delta^*\right) + j\frac{L_{\max}}{\varepsilon S}\left(\delta^0 + \Delta^*\right)\right)\right)$$

$$\geq \frac{\delta^0 L_-}{\varepsilon}\min_{j\in[n]}\left(\left(\sum_{i=1}^j \frac{1}{\tau_i}\right)^{-1}\left(\frac{L_{\max}}{\varepsilon}\left(\delta^0 + \Delta^*\right) + j\right)\right).$$

Finally, for $S = 1$, we have

$$T_{\text{SGD}}(S, [\tau_i]_{i=1}^n) = \frac{\delta^0 L_- L_{\max}}{\varepsilon^2}\left(\delta^0 + \Delta^*\right)\min_{j\in[n]}\left(\left(\sum_{i=1}^j \frac{1}{\tau_i}\right)^{-1}(1 + j)\right)$$

$$= \frac{\delta^0 L_-}{\varepsilon}\min_{j\in[n]}\left(\left(\sum_{i=1}^j \frac{1}{\tau_i}\right)^{-1}\left(\frac{L_{\max}}{\varepsilon}\left(\delta^0 + \Delta^*\right) + \frac{L_{\max}}{\varepsilon}\left(\delta^0 + \Delta^*\right)j\right)\right)$$

$$\geq \frac{\delta^0 L_-}{2\varepsilon}\min_{j\in[n]}\left(\left(\sum_{i=1}^j \frac{1}{\tau_i}\right)^{-1}\left(\frac{L_{\max}}{\varepsilon}\left(\delta^0 + \Delta^*\right) + j\right)\right)$$

because we assume $\frac{L_{\max}}{\varepsilon}\left(\delta^0 + \Delta^*\right) > \frac{1}{2}$. Therefore, an optimal choice is $S^* = \frac{L_{\max}}{\varepsilon}\left(\delta^0 + \Delta^*\right)$. $\quad\square$

# I   Setup of the Experiments from Section A.1

We consider the optimization problem (1) with nonconvex quadratic functions. The matrices and vectors defining the objective functions $f_i$ are generated using Algorithm 9 with $m = 10000$, $d = 1000$, $\lambda = 1\mathrm{e}{-6}$, and $s = 10$. The output is used to construct

$$f_i(x) = \frac{1}{2} x^\top \mathbf{A}_i x - b_i^\top x \quad \forall x \in \mathbb{R}^d, \ \forall i \in [m].$$

---

**Algorithm 9** Quadratic optimization task generation

---

1: **Parameters:** number of functions $m$, dimension $d$, regularizer $\lambda$, noise scale $s$
2: **for** $i = 1, \ldots, m$ **do**
3:     Generate random noises $\nu_i^s = 1 + s\xi_i^s$ and $\nu_i^b = s\xi_i^b$, i.i.d. $\xi_i^s, \xi_i^b \sim \mathcal{N}(0, 1)$
4:     Let $b_i = \frac{\nu_i^s}{4}(-1 + \nu_i^b, 0, \cdots, 0) \in \mathbb{R}^d$
5:     Take the initial tridiagonal matrix

$$\mathbf{A}_i = \frac{\nu_i^s}{4} \begin{pmatrix} 2 & -1 & & 0 \\ -1 & \ddots & \ddots & \\ & \ddots & \ddots & -1 \\ 0 & & -1 & 2 \end{pmatrix} \in \mathbb{R}^{d \times d}$$

6: **end for**
7: Take the mean of matrices $\mathbf{A} = \frac{1}{m} \sum_{i=1}^{m} \mathbf{A}_i$
8: Find the minimum eigenvalue $\lambda_{\min}(\mathbf{A})$
9: **for** $i = 1, \ldots, m$ **do**
10:     Update matrix $\mathbf{A}_i = \mathbf{A}_i + (\lambda - \lambda_{\min}(\mathbf{A}))\mathbf{I}$
11: **end for**
12: Take starting point $x^0 = (\sqrt{d}, 0, \cdots, 0)$
13: **Output:** matrices $\mathbf{A}_1, \cdots, \mathbf{A}_m$, vectors $b_1, \cdots, b_m$, starting point $x^0$

---

# J Lower bound

## J.1 Time multiple oracles protocol

The classical lower bound frameworks [Nemirovskij and Yudin, 1983, Carmon et al., 2020, Arjevani et al., 2022, Nesterov, 2018] are not convenient in the analysis of parallel algorithms since they are designed to estimate lower bounds on *iteration complexities*. In order to obtain *time complexity* lower bounds, we use the framework by Tyurin and Richtárik [2023]. Let us briefly explain the main idea. A more detailed explanation can be found in [Tyurin and Richtárik, 2023][Sections 3-6].

We start by introducing an appropriate oracle for our setup:

$$O_\tau : \underbrace{\mathbb{R}_{\geq 0}}_{\text{time}} \times \underbrace{\mathbb{R}^d}_{\text{point}} \times \underbrace{(\mathbb{R}_{\geq 0} \times \mathbb{R}^d \times \{0,1\})}_{\text{input state}} \to \underbrace{(\mathbb{R}_{\geq 0} \times \mathbb{R}^d \times \{0,1\})}_{\text{output state}} \times \mathbb{R}^d$$

such that $\quad O_\tau(t, x, (s_t, s_x, s_q)) = \begin{cases} ((t, x, 1), & 0), & s_q = 0, \\ ((s_t, s_x, 1), & 0), & s_q = 1, t < s_t + \tau, \\ ((0,0,0), & \nabla f_j(s_x)), & s_q = 1, t \geq s_t + \tau, \end{cases}$ (29)

where $j \sim \text{Uniform}([m])$, i.e., $j$ is a random index sampled uniformly from the set $[m]$. We assume that all draws from $\text{Uniform}([m])$ are i.i.d..

Next, we define the *time multiple oracles protocol*, first introduced in [Tyurin and Richtárik, 2023].

---

**Protocol 10** Time Multiple Oracles Protocol

---

1: **Input:** function $f = \frac{1}{m} \sum_{i=1}^m f_i$, oracles and distributions $(O_1, ..., O_n) \in \mathcal{O}(f)$, algorithm $A \in \mathcal{A}$
2: $s_i^0 = 0$ for all $i \in [n]$
3: **for** $k = 0, \ldots, \infty$ **do**
4: $\quad (t^{k+1}, i^{k+1}, x^k) = A^k(g^1, \ldots, g^k),$ $\qquad\qquad\qquad\qquad\qquad \triangleright t^{k+1} \geq t^k$
5: $\quad (s_{i^{k+1}}^{k+1}, g^{k+1}) = O_{i^{k+1}}(t^{k+1}, x^k, s_{i^{k+1}}^k)$ $\qquad\qquad \triangleright s_j^{k+1} = s_j^k \quad \forall j \neq i^{k+1}$
6: **end for**

---

Let us explain the behavior of the protocol. At each iteration, the algorithm $A$ returns three outputs, based on the available information/gradients: time $t^{k+1}$, the index of a worker $i^{k+1}$, and a new point $x^k$. Depending on the current time $t^{k+1}$ and the state of the worker, three options are possible (see (29)). If $s_q = 0$, then the worker is idle. It then starts calculations at the point $x^k$, changes the state $s_q$ from 0 to 1, stores the point $x^k$ in $s_x$ (at which a new stochastic gradient should be calculated), and returns a zero vector. If $s_q = 0$ and $t^{k+1} < s_t + \tau$, then the worker is still calculating a stochastic gradient. It does not change the state and returns a zero vector because the computation has not finished yet. If $s_q = 0$ and $t^{k+1} \geq s_t + \tau$, the worker can finally return a stochastic gradient at $s_x$ because sufficient time has passed since the worker was idle ($s_q = 0$). Note that with this oracle, the algorithm will never receive the first stochastic gradient before time $\tau$ (assuming that all oracles have the same processing time $\tau$; in general, we will assume that the processing times are different).

In the setting considered in this work, there are $n$ oracles that can do calculations in parallel, and an algorithm orchestrates their work. Let the processing times of the oracles be equal to $\tau_1, \ldots, \tau_n$. A reasonable strategy would be to call each oracle with $t^k = 0$, then to call the fastest worker with $t^k = \min_{i \in [n]} \tau_i$ to get the first stochastic gradients as soon as possible, then to call this worker again with $t^k = \min_{i \in [n]} \tau_i$ to request calculation of the next stochastic gradient, and so on. One unusual thing about this protocol is that the algorithm controls the time. The oracle is designed to force the algorithm to increase the time; otherwise, the algorithm would not receive new information about the function.

Our goal will be to bound the complexity measure $\mathfrak{m}_{\text{time}}(\mathcal{A}, \mathcal{F})$, defined as

$$\mathfrak{m}_{\text{time}}(\mathcal{A}, \mathcal{F}) := \inf_{A \in \mathcal{A}} \sup_{f \in \mathcal{F}} \sup_{\{O_i\} \in \mathcal{O}(f)} \inf \left\{ t \geq 0 \,\middle|\, \mathbb{E}\left[ \inf_{k \in S_t} \left\| \nabla f(x^k) \right\|^2 \right] \leq \varepsilon \right\},$$

$$S_t := \left\{ k \in \mathbb{N}_0 \,\middle|\, t^k \leq t \right\},$$ (30)

where the sequences $t^k$ and $x^k$ are generated by Protocol 10. Hence, unlike the classical approach, where the lower bounds are obtained for the minimum number of iterations required to find an $\varepsilon$–stationary point, we seek to find the minimum *time* needed to get an $\varepsilon$–stationary point.

We consider a standard for our setup class of functions [Fang et al., 2018]:

**Definition 26** (Function Class $\mathcal{F}^m_{\delta^0, L_+}$). We say that $f \in \mathcal{F}^m_{\delta^0, L_+}$ if it is $\delta^0$-bounded, i.e., $f(0) - \inf_{x \in \mathbb{R}^d} f(x) \leq \delta^0$, and

$$f(x) = \frac{1}{m} \sum_{i=1}^m f_i(x),$$

where the functions $f_i \,:\, \mathbb{R}^d \to \mathbb{R}$ are differentiable and satisfy

$$\frac{1}{m} \sum_{i=1}^m \|\nabla f_i(x) - \nabla f_i(y)\|^2 \leq L_+^2 \|x - y\|^2 \quad \forall x, y \in \mathbb{R}^d.$$

Next, we define the class of algorithms we will analyze.

**Definition 27** (Algorithm Class $\mathcal{A}_{\mathrm{zr}}$). Let us consider Protocol 10. We say that a sequence of mappings $A = \{A^k\}_{k=0}^\infty$ is a *zero-respecting algorithm*, if

1. $A^k \,:\, \underbrace{\mathbb{R}^d \times \cdots \times \mathbb{R}^d}_{k \text{ times}} \to \mathbb{R}_{\geq 0} \times \mathbb{N} \times \mathbb{R}^d$ for all $k \geq 1$ and $A^0 \in \mathbb{R}_{\geq 0} \times \mathbb{N} \times \mathbb{R}^d$.

2. For all $k \geq 1$ and $g^1, \ldots, g^k \in \mathbb{R}^d$, $t^{k+1} \geq t^k$, where $t^{k+1}$ and $t^k$ are defined as $(t^{k+1}, \ldots) = A^k(g^1, \ldots, g^k)$ and $(t^k, \ldots) = A^{k-1}(g^1, \ldots, g^{k-1})$.

3. $\mathrm{supp}\left(x^k\right) \subseteq \bigcup_{j=1}^k \mathrm{supp}\left(g^j\right)$ for all $k \in \mathbb{N}_0$, where $\mathrm{supp}(x) := \{i \in [d] \,|\, x_i \neq 0\}$.

We denote the set of all algorithms with these properties as $\mathcal{A}_{\mathrm{zr}}$.

In the above definition, property 1 defines the domain of the mappings $A^k$, and property 2 ensures that our algorithm does not "cheat" and does not "travel into the past": the time can only go forward (see [Tyurin and Richtárik, 2023][Section 4, Definition 4.1]). Property 3 is a standard assumption for zero-respecting algorithms [Arjevani et al., 2022] that is satisfied by virtually all algorithms, including Adam [Kingma and Ba, 2015], SGD, PAGE [Li et al., 2021] and Asynchronous SGD.

It remains to define an oracle class for our problem that employs oracles from (29). We design oracles that emulate the real behavior of the workers.

**Definition 28** (Computation Oracle Class $\mathcal{O}_{\tau_1, \ldots, \tau_n}$). For any $f \in \mathcal{F}^m_{\delta^0, L_+}$, the oracle class $\mathcal{O}_{\tau_1, \ldots, \tau_n}$ returns oracles $O_i = O_{\tau_i}$, $i \in [n]$, where the mappings $O_{\tau_i}$ are defined in (29).

## J.2 The "worst case" function in the nonconvex world

The analysis uses a standard function, commonly employed to derive lower bounds in the nonconvex regime. First, let us define

$$\mathrm{prog}(x) := \max\{i \geq 0 \,|\, x_i \neq 0\} \quad (x_0 \equiv 1).$$

Our choice of the underlying function $F$ follows the construction introduced in Carmon et al. [2020], Arjevani et al. [2022]: for any $T \in \mathbb{N}$, define

$$F_T(x) := -\Psi(1)\Phi(x_1) + \sum_{i=2}^T \left[\Psi(-x_{i-1})\Phi(-x_i) - \Psi(x_{i-1})\Phi(x_i)\right], \tag{31}$$

where

$$\Psi(x) = \begin{cases} 0, & x \leq 1/2, \\ \exp\left(1 - \frac{1}{(2x-1)^2}\right), & x \geq 1/2, \end{cases} \quad \text{and} \quad \Phi(x) = \sqrt{e} \int_{-\infty}^x e^{-\frac{1}{2}t^2} \, dt.$$

Throughout the proof, we only rely on the following properties of the function:

**Lemma 1** (Carmon et al. [2020], Arjevani et al. [2022]). *The function $F_T$ satisfies:*

1. $F_T(0) - \inf_{x \in \mathbb{R}^T} F_T(x) \leq \Delta^0 T$, *where* $\Delta^0 = 12$.

2. *The function $F_T$ is $l_1$–smooth, where* $l_1 = 152$.

3. *For all* $x \in \mathbb{R}^T$, $\|\nabla F_T(x)\|_\infty \leq \gamma_\infty$, *where* $\gamma_\infty = 23$.

4. *For all* $x \in \mathbb{R}^T$, $\mathrm{prog}(\nabla F_T(x)) \leq \mathrm{prog}(x) + 1$.

5. *For all* $x \in \mathbb{R}^T$, *if* $\mathrm{prog}(x) < T$, *then* $\|\nabla F_T(x)\| > 1$.

### J.3 The first lower bound

We are ready to present the main results of this section.

**Theorem 29.** *Let us consider Protocol 10. Without loss of generality, assume that $0 < \tau_1 \leq \cdots \leq \tau_n$ and take any $L_+, \delta^0, \varepsilon > 0$, and $m \in \mathbb{N}$ such that $\varepsilon < c_1 L_+ \delta^0$ and $\frac{\delta^0 L_+}{\varepsilon} > c_2 \sqrt{m}$. Then, for any algorithm $A \in \mathcal{A}_{\mathrm{zr}}$, there exists a function $f \in \mathcal{F}^m_{\delta^0, L_+}$ and computation oracles $(O_1, \ldots, O_n) \in \mathcal{O}_{\tau_1, \ldots, \tau_n}(f)$ such that $\mathbb{E}\left[\inf_{k \in S_t} \|\nabla f(x^k)\|^2\right] > \varepsilon$, where $S_t := \{k \in \mathbb{N}_0 \,|\, t^k \leq t\}$ and*

$$t = c_3 \times \frac{\delta^0 L_+}{\sqrt{m}\varepsilon} \min_{j \in [n]} \left[\left(\sum_{i=1}^{j} \frac{1}{\tau_i}\right)^{-1} (m+j)\right].$$

*The quantities $c_1, c_2$, and $c_3$ are universal constants. The sequences $x^k$ and $t^k$ are defined in Protocol 10.*

*Proof.* **(Step 1: Construction of a hard problem)**
We start our proof by constructing an appropriate function $f \in \mathcal{F}^m_{\delta^0, L_+}$. Let us fix any $T \geq \mathbb{N}$ and define $f_i : \mathbb{R}^T \to \mathbb{R}$ such that

$$f_1 := \frac{\sqrt{m}L_+\lambda^2}{l_1} F_T\left(\frac{x}{\lambda}\right)$$

for all $x \in \mathbb{R}^T$, and $f_i(x) = 0$ for all $i \in \{2, \ldots, m\}$ and $x \in \mathbb{R}^T$. Essentially, all information about the function $f = \frac{1}{m}\sum_{i=1}^{m} f_i$ is in the first function. Note that

$$\frac{1}{m}\sum_{i=1}^{m} \|\nabla f_i(x) - \nabla f_i(y)\|^2 = \frac{1}{m}\|\nabla f_1(x) - \nabla f_1(y)\|^2$$

$$= \frac{1}{m}\left\|\frac{\sqrt{m}L_+\lambda}{l_1}\nabla F_T\left(\frac{x}{\lambda}\right) - \frac{\sqrt{m}L_+\lambda}{l_1}\nabla F_T\left(\frac{y}{\lambda}\right)\right\|^2$$

$$= \frac{L_+^2\lambda^2}{l_1^2}\left\|\nabla F_T\left(\frac{x}{\lambda}\right) - \nabla F_T\left(\frac{y}{\lambda}\right)\right\|^2.$$

Then, using Lemma 1, we have

$$\frac{1}{m}\sum_{i=1}^{m} \|\nabla f_i(x) - \nabla f_i(y)\|^2 \leq L_+^2\lambda^2\left\|\frac{x}{\lambda} - \frac{y}{\lambda}\right\|^2 = L_+^2\|x - y\|^2. \tag{32}$$

Taking

$$T = \left\lfloor \frac{\sqrt{m}\delta^0 l_1}{L_+\lambda^2\Delta^0}\right\rfloor,$$

we ensure that

$$f(0) - \inf_{x \in \mathbb{R}^T} f(x) = \frac{1}{m}\left(\frac{\sqrt{m}L_+\lambda^2}{l_1}F_T(0) - \inf_{x \in \mathbb{R}^T}\frac{\sqrt{m}L_+\lambda^2}{l_1}F_T(x)\right)$$

$$= \frac{L_+\lambda^2}{\sqrt{m}l_1}\left(F_T(0) - \inf_{x\in\mathbb{R}^T} F_T(x)\right) \leq \frac{L_+\lambda^2\Delta^0 T}{\sqrt{m}l_1} \leq \delta^0, \qquad (33)$$

where in the inequalities we use Lemma 1 and the choice of $T$. Now, inequalities (32) and (33) imply that $f = \frac{1}{m}\sum_{i=1}^m f_i \in \mathcal{F}_{\delta^0, L_+}^m$, and hence, using Lemma 1 again, we get

$$\inf_{k\in S^t}\left\|\nabla f(x^k)\right\|^2 = \inf_{k\in S^t}\left\|\frac{1}{m}\times\frac{\sqrt{m}L_+\lambda}{l_1}\nabla F_T\left(\frac{x^k}{\lambda}\right)\right\|^2$$
$$= \frac{L_+^2\lambda^2}{ml_1^2}\inf_{k\in S^t}\left\|\nabla F_T\left(\frac{x}{\lambda}\right)\right\|^2 > \frac{L_+^2\lambda^2}{ml_1^2}\inf_{k\in S^t}\mathbb{1}\left[\mathrm{prog}(x^k) < T\right].$$

Let us take

$$\lambda = \frac{2l_1\sqrt{m}\sqrt{\varepsilon}}{L_+}.$$

Then

$$\inf_{k\in S^t}\left\|\nabla f(x^k)\right\|^2 > 4\varepsilon\inf_{k\in S^t}\mathbb{1}\left[\mathrm{prog}(x^k) < T\right] \qquad (34)$$

and

$$T = \left\lfloor \frac{\delta^0 L_+}{4l_1\Delta^0\sqrt{m}\varepsilon} \right\rfloor.$$

The last inequality means that while $\mathrm{prog}(x^k) < T$ for all $k \in S^t$, all gradients are large. The function $F_T$ is a *zero-chain*: due to Lemma 1, we know that $\mathrm{prog}(\nabla F_T(x)) \leq \mathrm{prog}(x) + 1$ for all $x \in \mathbb{R}^T$. This implies that we can discover at most one new non-zero coordinate by calculating the gradient of the function $\nabla f_1$. Since the algorithm $A \in \mathcal{A}_{\mathrm{zr}}$ is zero-respecting, by definition it cannot return a point $x^k$ with progress greater than that of the vectors returned by the oracles. In view of this, it is necessary to calculate the gradient of $f_1$ at least $T$ times to get $\mathrm{prog}(x^k) \geq T$.

The gradient of $f_1$ can be calculated if and only if $j = 1$, where $j \sim \mathrm{Uniform}([m])$ (see (29)). Consider worker $i$ and define $\eta_i^1$ to be the number of draws from $\mathrm{Uniform}([m])$ until the index $j = 1$ is sampled. Clearly, $\eta_i^1$ is a Geometric random variable with parameter $\mathbb{P}(j = 1) = \frac{1}{m}$. Recall that the workers can do the computations in parallel, and by the design of the oracles, worker $i$ needs at least $\tau_i\eta_i^1$ seconds to calculate $\eta_i^1$ stochastic gradients. Hence, it is impossible to calculate $\nabla f_1$ before the time

$$\min_{i\in[n]}\tau_i\eta_i^1.$$

Once the algorithm calculates $\nabla f_1$ for the first time, it needs to do so at least $T - 1$ times more to achieve $\mathrm{prog}(x^k) \geq T$. Thus, one should wait at least

$$\sum_{k=1}^T\min_{i\in[n]}\tau_i\eta_i^k$$

seconds, where $\eta_i^k \overset{\mathrm{i.i.d.}}{\sim} \mathrm{Geometric}(1/m)$. We can conclude that

$$\mathbb{P}\left(\inf_{k\in S^t}\mathbb{1}\left[\mathrm{prog}(x^k) < T\right] = 0\right) \leq \mathbb{P}\left(\sum_{k=1}^T\min_{i\in[n]}\tau_i\eta_i^k \leq t\right). \qquad (35)$$

**(Step 2: The Chernoff Method)**
The theorem's proof is now reduced to the analysis of the concentration of $\sum_{k=1}^T\min_{i\in[n]}\tau_i\eta_i^k$. Using the Chernoff method, for all $s > 0$, we have

$$\mathbb{P}\left(\sum_{k=1}^T\min_{i\in[n]}\tau_i\eta_i^k \leq t\right) = \mathbb{P}\left(-s\sum_{k=1}^T\min_{i\in[n]}\tau_i\eta_i^k \geq -st\right)$$
$$= \mathbb{P}\left(e^{-s\sum_{k=1}^T\min_{i\in[n]}\tau_i\eta_i^k} \geq e^{-st}\right)$$

$$\leq e^{st}\mathbb{E}\left[\exp\left(-s\sum_{k=1}^{T}\min_{i\in[n]}\tau_i\eta_i^k\right)\right].$$

Independence gives

$$\mathbb{P}\left(\sum_{k=1}^{T}\min_{i\in[n]}\tau_i\eta_i^k\leq t\right)\leq e^{st}\prod_{k=1}^{T}\mathbb{E}\left[\exp\left(-s\min_{i\in[n]}\tau_i\eta_i^k\right)\right]$$

$$\stackrel{\text{i.i.d.}}{=}e^{st}\left(\mathbb{E}\left[\exp\left(-s\min_{i\in[n]}\tau_i\eta_i^1\right)\right]\right)^{T}. \tag{36}$$

Let us consider the term in the last bracket separately. For a fixed $t'>0$, we have

$$\mathbb{E}\left[\exp\left(-s\min_{i\in[n]}\tau_i\eta_i^1\right)\right] \tag{37}$$

$$=\quad\mathbb{E}\left[\max_{i\in[n]}\exp\left(-s\tau_i\eta_i^1\right)\right]$$

$$=\quad\mathbb{E}\left[\max_{i\in[n]}\left(\mathbb{1}\left[\tau_i\eta_i^1\leq t'\right]\exp\left(-s\tau_i\eta_i^1\right)+\left(1-\mathbb{1}\left[\tau_i\eta_i^1\leq t'\right]\right)\exp\left(-s\tau_i\eta_i^1\right)\right)\right]$$

$$\leq\quad\mathbb{E}\left[\max_{i\in[n]}\left(\mathbb{1}\left[\tau_i\eta_i^1\leq t'\right]+\left(1-\mathbb{1}\left[\tau_i\eta_i^1\leq t'\right]\right)\exp\left(-st'\right)\right)\right]$$

$$=\quad\exp\left(-st'\right)+\left(1-\exp\left(-st'\right)\right)\mathbb{E}\left[\max_{i\in[n]}\left(\mathbb{1}\left[\tau_i\eta_i^1\leq t'\right]\right)\right]. \tag{38}$$

We now consider the last term. Due to the independence, we have

$$\mathbb{E}\left[\max_{i\in[n]}\left(\mathbb{1}\left[\tau_i\eta_i^1\leq t'\right]\right)\right]=1-\prod_{i=1}^{n}\mathbb{P}\left(\tau_i\eta_i^1>t'\right)=1-\prod_{i=1}^{n}(1-p)^{\left\lfloor\frac{t'}{\tau_i}\right\rfloor},$$

where we use the cumulative distribution function of a geometric random variable and temporarily define $p:=\frac{1}{m}$. Using Lemma 3, we get

$$\mathbb{E}\left[\max_{i\in[n]}\left(\mathbb{1}\left[\tau_i\eta_i^1\leq t'\right]\right)\right]\leq p\sum_{i=1}^{n}\left\lfloor\frac{t'}{\tau_i}\right\rfloor. \tag{39}$$

Let us take

$$t'=\frac{1}{8}\times\min_{j\in[n]}\left(\sum_{i=1}^{j}\frac{1}{\tau_i}\right)^{-1}\left(\frac{1}{p}+j\right)=\frac{1}{8}\times\min_{j\in[n]}g(j),$$

where $g(j):=\left(\sum_{i=1}^{j}\frac{1}{\tau_i}\right)^{-1}\left(\frac{1}{p}+j\right)$ for all $j\in[n]$ and assume that $j^*$ is the largest index such that $\min_{j\in[n]}g(j)=g(j^*)$. Then, Lemma 4 gives

$$\tau_{j^*}\leq\min_{j\in[n]}g(j)<\tau_{j^*+1}, \tag{40}$$

where we let $\tau_{n+1}\equiv\infty$. Therefore, $t'<\tau_{j^*+1}$ and (39) gives

$$\mathbb{E}\left[\max_{i\in[n]}\left(\mathbb{1}\left[\tau_i\eta_i^1\leq t'\right]\right)\right]\leq p\sum_{i=1}^{n}\left\lfloor\frac{t'}{\tau_i}\right\rfloor=p\sum_{i=1}^{j^*}\left\lfloor\frac{t'}{\tau_i}\right\rfloor.$$

Using (40), we get $\frac{t'}{\tau_i}=\frac{\min_{j\in[n]}g(j)}{8\tau_i}\geq\frac{1}{8}$ for all $i\leq j^*$. Since $\lfloor x\rfloor\leq 2x-\frac{1}{4}$ for all $x\geq\frac{1}{8}$, we obtain

$$\mathbb{E}\left[\max_{i\in[n]}\left(\mathbb{1}\left[\tau_i\eta_i^1\leq t'\right]\right)\right]\leq p\sum_{i=1}^{j^*}\left(\frac{2t'}{\tau_i}-\frac{1}{4}\right)=2pt'\left(\sum_{i=1}^{j^*}\frac{1}{\tau_i}\right)-\frac{pj^*}{4}$$

$$= 2p \times \frac{1}{8} \left( \sum_{i=1}^{j^*} \frac{1}{\tau_i} \right)^{-1} \left( \frac{1}{p} + j^* \right) \times \left( \sum_{i=1}^{j^*} \frac{1}{\tau_i} \right) - \frac{pj^*}{4}$$

$$= \frac{p}{4} \left( \frac{1}{p} + j^* \right) - \frac{pj^*}{4} = \frac{1}{4}.$$

Substituting the last inequality to (38) gives

$$\mathbb{E} \left[ \exp \left( -s \min_{i \in [n]} \tau_i \eta_i^1 \right) \right] \leq \exp \left( -st' \right) + \frac{1}{4} (1 - \exp \left( -st' \right)).$$

We now take $s = 1/t'$ to obtain

$$\mathbb{E} \left[ \exp \left( -s \min_{i \in [n]} \tau_i \eta_i^1 \right) \right] \leq e^{-1} + \frac{1}{4} (1 - e^{-1}) \leq e^{-\frac{1}{2}}.$$

Substituting this inequality in (35) and (36) gives

$$\mathbb{P} \left( \inf_{k \in S^t} \mathbb{1} \left[ \text{prog}(x^k) < T \right] = 0 \right) \leq \mathbb{P} \left( \sum_{k=1}^{T} \min_{i \in [n]} \tau_i \eta_i^k \leq t \right) \leq e^{\frac{t}{t'} - \frac{T}{2}}.$$

Therefore,

$$\mathbb{P} \left( \inf_{k \in S^t} \mathbb{1} \left[ \text{prog}(x^k) < T \right] = 0 \right) \leq \rho$$

for all

$$t \leq \frac{1}{8} \min_{j \in [n]} \left[ \left( \sum_{i=1}^{j} \frac{1}{\tau_i} \right)^{-1} \left( \frac{1}{p} + j \right) \right] \left( \frac{T}{2} + \log \rho \right)$$

and $\rho > 0$. Using the bound on the probability with $\rho = \frac{1}{2}$ and (34), we finally conclude

$$\mathbb{E} \left[ \inf_{k \in S^t} \left\| \nabla f(x^k) \right\|^2 \right] > 4\varepsilon \mathbb{P} \left( \inf_{k \in S^t} \mathbb{1} \left[ \text{prog}(x^k) < T \right] = 1 \right) \geq 2\varepsilon$$

for all

$$t \leq \frac{1}{8} \min_{j \in [n]} \left[ \left( \sum_{i=1}^{j} \frac{1}{\tau_i} \right)^{-1} \left( \frac{1}{p} + j \right) \right] \left( \frac{T}{2} + \log \frac{1}{2} \right)$$

$$= \frac{1}{8} \min_{j \in [n]} \left[ \left( \sum_{i=1}^{j} \frac{1}{\tau_i} \right)^{-1} (m + j) \right] \left( \frac{1}{2} \left\lfloor \frac{\delta^0 L_+}{4 l_1 \Delta^0 \sqrt{m\varepsilon}} \right\rfloor + \log \frac{1}{2} \right).$$

It remains to use the conditions $\varepsilon < c_1 L \delta^0$ and $\frac{\delta^0 L_+}{\varepsilon} > c_2 \sqrt{m}$ from the theorem to finish the proof. $\qquad \square$

### J.4 The second lower bound

**Theorem 30.** *Let us consider Protocol 10. Without loss of generality, assume that $0 < \tau_1 \leq \cdots \leq \tau_n$ and take any $L_+, \delta^0, \varepsilon > 0$, and $m \in \mathbb{N}$ such that $\varepsilon < c_1 L_+ \delta^0$. Then, for any algorithm $A \in \mathcal{A}_{zr}$, there exists a function $f \in \mathcal{F}_{\delta^0, L_+}^m$ and computation oracles $(O_1, \ldots, O_n) \in \mathcal{O}_{\tau_1, \ldots, \tau_n}(f)$ such that $\mathbb{E} \left[ \inf_{k \in S_t} \left\| \nabla f(x^k) \right\|^2 \right] > \varepsilon$, where $S_t := \left\{ k \in \mathbb{N}_0 \,|\, t^k \leq t \right\}$ and*

$$t = c_2 \times \min_{j \in [n]} \left( \sum_{i=1}^{j} \frac{1}{\tau_i} \right)^{-1} (m + j).$$

*The quantities $c_1$ and $c_2$ are universal constants. The sequences $x^k$ and $t^k$ are defined in Protocol 10.*

*Proof.* We use the same construction as in the proof of Theorem 2 of Li et al. [2021]. Let us consider

$$f_i(x) := c \langle v_i, x \rangle + \frac{L_+}{2} \|x\|^2 \tag{41}$$

for all $x \in \mathbb{R}^T$, where $c \in \mathbb{R}$ and $v_i \in \mathbb{R}^T$, $i \in [n]$ are parameters that we define later. Then

$$\frac{1}{m} \sum_{i=1}^m \|\nabla f_i(x) - \nabla f_i(y)\|^2 \le L_+^2 \|x - y\|^2$$

for all $x, y \in \mathbb{R}^d$ and

$$f(0) - \inf_{x \in \mathbb{R}^T} f(x) = - \inf_{x \in \mathbb{R}^T} \left[ c \left\langle \frac{1}{m} \sum_{i=1}^m v_i, x \right\rangle + \frac{L_+}{2} \|x\|^2 \right] = \frac{c^2}{2L_+ m^2} \left\| \sum_{i=1}^m v_i \right\|^2 = \delta^0,$$

where we take

$$c := \sqrt{\frac{2L_+ m^2 \delta^0}{\left\| \sum_{i=1}^m v_i \right\|^2}}.$$

Thus, we have $f = \frac{1}{m} \sum_{i=1}^m f_i \in \mathcal{F}_{\delta^0, L_+}^m$. Now, let

$$v_1 = (\underbrace{1, \ldots, 1}_{\frac{T}{m}}, 0, \ldots, 0)^\top \in \mathbb{R}^T,$$

$$v_2 = (\underbrace{0, \ldots, 0}_{\frac{T}{m}}, \underbrace{1, \ldots, 1}_{\frac{T}{m}}, 0, \ldots, 0)^\top \in \mathbb{R}^T,$$

$$v_3 = (\underbrace{0, \ldots, 0}_{\frac{T}{m}}, \underbrace{0, \ldots, 0}_{\frac{T}{m}}, \underbrace{1, \ldots, 1}_{\frac{T}{m}}, 0, \ldots, 0)^\top \in \mathbb{R}^T$$

$$\ldots$$

$$v_n = (0, \ldots, 0, \underbrace{1, \ldots, 1}_{\frac{T}{m}})^\top \in \mathbb{R}^T.$$

and choose any $T \in \{sm \mid s \in \mathbb{N}\}$ (one can always take $T = m$). Then

$$c = \sqrt{\frac{2L_+ m^2 \delta^0}{T}}.$$

Let us fix some time $t > 0$ to be determined later and recall that the workers can calculate the stochastic gradients $\nabla f_i(x) = cv_i + L_+ x$ in parallel. Suppose that up to time $t$, fewer than $\frac{m}{2}$ stochastic gradients have been computed. Then, since the algorithm is a zero-respecting algorithm, it cannot have discovered more than $\frac{m}{2} \times \frac{T}{m} = \frac{T}{2}$ coordinates. Thus, at least $\frac{T}{2}$ coordinates are equal to 0 and

$$\left\| \nabla f(x^k) \right\|^2 = \left\| \frac{c}{m} \sum_{i=1}^m v_i + L_+ x^k \right\|^2 \ge \frac{c^2}{m^2} \times \frac{T}{2}.$$

for all $k \in \{k \in \mathbb{N}_0 \mid t^k \le t\}$. Therefore, substituting our choice of $c$ and using the assumptions from the theorem, we get

$$\left\| \nabla f(x^k) \right\|^2 \ge L_+ \delta^0 > 2\varepsilon. \tag{42}$$

It remains to find the time $t > 0$. The workers work in parallel, so in $t$ seconds they calculate at most

$$\sum_{i=1}^n \left\lfloor \frac{t}{\tau_i} \right\rfloor$$

stochastic gradients. Now, let

$$t = \frac{1}{8} \min_{j \in [n]} \left( \sum_{i=1}^{j} \frac{1}{\tau_i} \right)^{-1} (m + j),$$
(43)

and define $g(j) := \left( \sum_{i=1}^{j} \frac{1}{\tau_i} \right)^{-1} (m + j)$ for all $j \in [n]$. Assume that $j^*$ is the largest index such that $\min_{j \in [n]} g(j) = g(j^*)$. Using Lemma 4, we have

$$\sum_{i=1}^{n} \left\lfloor \frac{t}{\tau_i} \right\rfloor = \sum_{i=1}^{j^*} \left\lfloor \frac{t}{\tau_i} \right\rfloor.$$

Since $\frac{t}{\tau_i} \geq \frac{1}{8}$ for all $i \leq j^*$ and $\lfloor x \rfloor \leq 2x - \frac{1}{4}$ for all $x \geq \frac{1}{8}$, we obtain

$$\sum_{i=1}^{n} \left\lfloor \frac{t}{\tau_i} \right\rfloor \leq \sum_{i=1}^{j^*} \frac{2t}{\tau_i} - \frac{j^*}{4} = \frac{1}{4} \left( \sum_{i=1}^{j^*} \frac{1}{\tau_i} \right)^{-1} (m + j^*) \sum_{i=1}^{j^*} \frac{1}{\tau_i} - \frac{j^*}{4} = \frac{m}{4}.$$

Therefore, it is possible to calculate at most $\frac{m}{4}$ stochastic gradient in time (43) and we can finally conclude that inequality (42) holds for any time that is less than or equal (43). $\qquad \square$

# K   Useful Identities and Inequalities

**Lemma 2** ([Szlendak et al., 2021]). *It holds that $L_- \leq L_+$, $L_- \leq \frac{1}{m} \sum_{i=1}^m L_i$ and $L_+^2 - L_-^2 \leq L_\pm^2 \leq L_+^2 \leq \frac{1}{m} \sum_{i=1}^m L_i^2$.*

**Lemma 3.** *Consider a sequence $q_1, \ldots, q_n \in [0,1]$. Then*

$$1 - \sum_{m=1}^n q_m \leq \prod_{m=1}^n (1 - q_m).$$

*Proof.* We prove the result by induction. It is clearly true for $n = 1$: $1 - \sum_{m=1}^1 q_m = \prod_{m=1}^1 (1 - q_m)$. Now, assume that that it holds for $n-1$, meaning that

$$1 - \sum_{m=1}^{n-1} q_m \leq \prod_{m=1}^{n-1} (1 - q_m).$$

Multiplying both sides of the inequality by $1 - q_n \in [0,1]$ gives

$$\prod_{m=1}^n (1 - q_m) \geq (1 - q_n)\left(1 - \sum_{m=1}^{n-1} q_m\right) = 1 - \sum_{m=1}^{n-1} q_m - q_n + q_n \left(\sum_{m=1}^{n-1} q_m\right) \geq 1 - \sum_{m=1}^n q_m$$

since $q_m \in [0,1]$ for all $m \in [n]$. $\qquad\square$

**Theorem 31.** *Let us consider the equilibrium time mapping from Definition 3. Then*

*1. $t^*(S, [\tau_i]_{i=1}^n) \leq 2\tau_n \max\left\{\frac{S}{n}, 1\right\}.$*

*2. $t^*(S, [\tau_i]_{i=1}^n) \leq 2\tau_1 \max\left\{S, 1\right\}$*

*for all $S \geq 0$, $\tau_i \in [0, \infty]$ for all $i \in [n]$, and $\tau_1 \leq \cdots \leq \tau_n$.*

**Remark 32.** *Assume that $\tau_1 = \cdots = \tau_{n-1} = \tau$ and $\tau_n \to \infty$, then*

$$\lim_{\tau_n \to \infty} t^*(S, [\tau_i]_{i=1}^n) = t^*(S, [\tau_i]_{i=1}^{n-1}) = \tau\left(\frac{S}{n-1} + 1\right) \leq 2\tau \max\left\{\frac{S}{n-1}, 1\right\},$$

$$\lim_{\tau_n \to \infty} 2\tau_n \max\left\{\frac{S}{n}, 1\right\} = \infty,$$

*and*

$$\lim_{\tau_n \to \infty} 2\tau_1 \max\left\{S, 1\right\} = 2\tau \max\left\{S, 1\right\}.$$

*Thus, $t^*(S, [\tau_i]_{i=1}^n)$ can be arbitrarily smaller than $2\tau_n \max\left\{\frac{S}{n}, 1\right\}$ and $2\tau_1 \max\left\{S, 1\right\}$. This implies that our new complexities (9) and (10) can be arbitrarily better than $T_{\text{Soviet PAGE}}$ and $T_{\text{Hero PAGE}}$.*

**Lemma 4.** *Consider a sequence $0 < \tau_1 \leq \ldots \leq \tau_n$ and fix some $S > 0$. For all $j \in [n]$, define*

$$g(j) := \left(\sum_{i=1}^j \frac{1}{\tau_i}\right)^{-1} (S + j).$$

*1. Let $j_{\max}^*$ be the largest index such that $\min_{j \in [n]} g(j) = g(j_{\max}^*)$. For $j_{\max}^* < n$, we have*

$$\min_{j \in [n]} g(j) < \tau_{(j_{\max}^* + 1)}.$$

*2. Let $j^*$ be any index such that $\min_{j \in [n]} g(j) = g(j^*)$. For $j^* < n$, we have*

$$\min_{j \in [n]} g(j) \leq \tau_{(j^* + 1)}.$$

3. Let $j^*_{\min}$ be the smallest index such that $\min_{j\in[n]} g(j) = g(j^*_{\min})$. Then

$$\tau_{j^*_{\min}} < \min_{j\in[n]} g(j).$$

4. Let $j^*$ be any index such that $\min_{j\in[n]} g(j) = g(j^*)$. Then

$$\tau_{j^*} \le \min_{j\in[n]} g(j).$$

*Proof.* We first prove the first inequality. Suppose that $j^*_{\max} < n$. Then $g(j^*_{\max}) < g(j^*_{\max} + 1)$, meaning that

$$\left(\sum_{i=1}^{j^*_{\max}} \frac{1}{\tau_i}\right)^{-1} (S + j^*_{\max}) < \left(\sum_{i=1}^{j^*_{\max}+1} \frac{1}{\tau_i}\right)^{-1} (S + j^*_{\max} + 1).$$

This implies the following series of inequalities:

$$\left(\sum_{i=1}^{j^*_{\max}+1} \frac{1}{\tau_i}\right) (S + j^*_{\max}) < \left(\sum_{i=1}^{j^*_{\max}} \frac{1}{\tau_i}\right) (S + j^*_{\max} + 1)$$

$$\Leftrightarrow \quad \frac{1}{\tau_{(j^*_{\max}+1)}} (S + j^*_{\max}) < \sum_{i=1}^{j^*_{\max}} \frac{1}{\tau_i}$$

$$\Leftrightarrow \quad \tau_{(j^*_{\max}+1)} > \left(\sum_{i=1}^{j^*_{\max}} \frac{1}{\tau_i}\right)^{-1} (S + j^*_{\max}) = g(j^*_{\max}) = \min_{j\in[n]} g(j).$$

The proof of the second inequality is the same, but with non-strict inequalities. To prove the third inequality, first suppose that $j^*_{\min} = 1$. Then $g(j^*_{\min}) = \tau_{j^*_{\min}}(S + 1) > \tau_{j^*_{\min}}$ as needed. On the other hand, $j^*_{\min} > 1$ implies

$$g(j^*_{\min}) < g(j^*_{\min} - 1)$$

$$\Leftrightarrow \left(\sum_{i=1}^{j^*_{\min}} \frac{1}{\tau_i}\right)^{-1} (S + j^*_{\min}) < \left(\sum_{i=1}^{j^*_{\min}-1} \frac{1}{\tau_i}\right)^{-1} (S + j^*_{\min} - 1)$$

$$\Leftrightarrow \left(\sum_{i=1}^{j^*_{\min}-1} \frac{1}{\tau_i}\right) (S + j^*_{\min}) < \left(\sum_{i=1}^{j^*_{\min}} \frac{1}{\tau_i}\right) (S + j^*_{\min} - 1)$$

$$\Leftrightarrow \sum_{i=1}^{j^*_{\min}} \frac{1}{\tau_i} < \frac{1}{\tau_{j^*_{\min}}} (S + j^*_{\min})$$

$$\Leftrightarrow \tau_{j^*_{\min}} < \left(\sum_{i=1}^{j^*_{\min}} \frac{1}{\tau_i}\right)^{-1} (S + j^*_{\min}) = \min_{j\in[n]} g(j).$$

The proof of the fourth inequality is the same, but with non-strict inequalities. $\square$

