# OpenReview forum: "Freya PAGE: First Optimal Time Complexity for Large-Scale Nonconvex Finite-Sum Optimization with Heterogeneous Asynchronous Computations"
_NeurIPS.cc/2024/Conference — NeurIPS 2024 poster_

### Official Review · Reviewer_wr2A · 2024-06-27

**Soundness:** 3
**Presentation:** 3
**Contribution:** 3
**Rating:** 6
**Confidence:** 3

**Summary:**

In this study, the authors developed a new distributed optimization algorithm, Freya Page method, for the non-convex finite-sum optimization problem. The authors provided the corresponding iteration and time complexity bounds. In addition, a lower bound on the time complexity is also provided.

**Strengths:**

The paper is well-written and easy to follow. The developed Algorithms 2-3 are novel and lead to useful time complexity bounds. The results should be interesting to audiences in distributed optimization and machine learning fields. Due to the time limit, I did not check the proofs in the appendix. But I feel that the proofs should be correct, except that of Theorem 10, which I have a concern about.

**Weaknesses:**

I did not see major weaknesses. But I am a little confused by the lower bound in Theorem 10 and the claims after the theorem. I have also included a few comments for the authors' consideration.

**Questions:**

- I wonder if the results can be extended to the case when the computation time for each gradient $\nabla f_j$ is different for different $j$. In other words, can we replace $\tau_{i}$ in Line 31 with $\tau_{i, j}$, which may be different for different $j$?

- Line 118: It would be better if the authors can provide more details on the choice of inverse proportion to $\tau_i$. I feel that this choice can guarantee that the worst-case running time is the same for all workers?

- In Section 1, the authors claimed that Theorem 4 allows $\tau_i = \infty$ for all $i$. But I feel that the authors are referring to Theorem 5? Also, I think Theorem 5 (as well as Theorems 1-2) requires $\tau_i$ to be finite for some $i$ so that $t^*$ is finite? It seems that this is supported by Example 3, where the current analysis is not able to provide an upper bound on the time complexity.

- For the time complexity in Theorem 4, I think the authors are computing the *expected* time complexity. This information is not mentioned in the paper and I feel it might be better to add a formal definition of the expected time complexity in the paper.

- It seems that the authors defined the notion of $t^*(S)$ in Definition 3 but did not used it in the following theorems.

- Line 200: I am not sure why the monotony of the two terms enable the application of the bi-section method.

- Line 294: I am a little confused when the authors mentioned that (10) is less than or equal to (12). It seems that (12) should be a lower bound on the complexity (up to a constant) and the achieved time complexity (10) should not be smaller than it?

**Limitations:**

See my comments in the Questions section.

---

> ### Author Rebuttal · Authors · 2024-08-04
>
> We appreciate your comments and are grateful for highlighting the positive aspects of our work. We will now proceed to address the questions raised and provide clarifications.
>
> > I wonder if the results can be extended to the case when the computation time for each gradient $\nabla f_j$ is different for different $j$. In other words, can we replace $\tau_i$ in Line 31 with $\tau_{i,j}$, which may be different for different $j$?
>
> If the processing times varied across different data points, one could always set $\tau_i = \max_j \tau_{i,j}$, and our theory would still be applicable without requiring any adjustments. However, there are probably better strategies than this, which we plan to investigate in future work.
>
> > Line 118: It would be better if the authors can provide more details on the choice of inverse proportion to $\tau_i$. I feel that this choice can guarantee that the worst-case running time is the same for all workers?
>
> The idea is the same as that in the proof of Theorem 2 in the appendix, which shows that a batch of $m$ data points can be processed in time (3). We agree that this should be stated more clearly in the main part of the paper and will make the necessary adjustments in the revised version.
>
> > In Section 1, the authors claimed that Theorem 4 allows $\tau_i=\infty$ for all $i$. But I feel that the authors are referring to Theorem 5? Also, I think Theorem 5 (as well as Theorems 1-2) requires $\tau_i$ to be finite for some $i$ so that $t^*$ is finite?
>
> Theorem 4 establishes the iteration complexity, and the total number of iterations to find an $\varepsilon$-stationary point is independent of the processing times. Therefore, the upper bounds can indeed be infinite, and the result still holds.
>
> If we assumed that the processing times of workers are *exactly* equal to $[\tau_i],$ then indeed we would have to assume $\tau_i$ to be finite for some $i.$ However, throughout the paper, we only assume that $\tau_i$ provides an upper bound on the processing times.
>
> For instance, consider an example where all workers have processing times equal to $1$ in the first iteration. Then, in the second iteration, the first worker turns off, meaning its processing time is $\infty.$ All other workers continue to have processing times equal to $1.$ In the third iteration, the first worker turns on with a processing time equals $1,$ but the second worker turns off, and so on. In this mathematical example, all workers turn off and turn on in a cyclic manner. Clearly, $\tau_i = \infty$ for all $i \in [n]$ because there is at least one iteration in which worker $i$ is turned off. However, in every iteration, there are at least $n - 1$ workers with processing times of $1.$
> This example is closely related to Section 4.4, where we allow the upper bound $\tau_i$ on processing times to be _dynamic_.
>
> > For the time complexity in Theorem 4, I think the authors are computing the expected time complexity. This information is not mentioned in the paper and I feel it might be better to add a formal definition of the expected time complexity in the paper.
>
> Theorem 4 states the iteration complexity, so we believe the reviewer is referring to the next result. Thank you for the suggestion. Indeed, this is an oversight on our part, and we will include a formal definition in the revised version.
>
> > It seems that the authors defined the notion of $t^*(S)$ in Definition 3 but did not used it in the following theorems.
>
> This shorthand notation is used in the text (see e.g. lines 183, 184) and in the proofs in the appendix.
>
> > Line 200: I am not sure why the monotony of the two terms enable the application of the bi-section method.
>
> Let us consider the problem $\min_{S \in N} h(S) + g(S),$ where $h(S) \geq 0$ is non-decreasing, $g(S) \geq 0$ is non-increasing, and $S^*$ is a solution. In order to solve it up to a constant factor (in the paper, we forgot to stress that the solution will be up to a constant factor), we can instead solve the equation $h(\bar{S}) - g(\bar{S}) = 0$ (a solution exists because $g(1) \geq h(1)$ and $h(S) \to \infty $ when $S \to \infty $ in our setting). Since $h(S) - g(S)$ is non-decreasing, $\bar{S}$ can be found using the bisection method.
>
> If $\bar{S} > S^*,$ then $h(S^*) + g(S^*) \geq g(\bar{S})$ because $g$ is non-increasing and $h(S^*) \geq 0.$ Then
> $h(S^*) + g(S^*) \geq \frac{1}{2} \left(h(\bar{S}) + g(\bar{S})\right)$ because $\bar{S}$ is a solution of $h(\bar{S}) = g(\bar{S}).$
>
> If $\bar{S} < S^*,$ then $h(S^*) + g(S^*) \geq h(\bar{S}) = \frac{1}{2} \left(h(\bar{S}) + g(\bar{S})\right).$
>
> Hence, $\bar{S}$ is a minimizer of $h(S) + g(S)$ up to the constant factor of $2.$ It remains to round $\bar{S}$ to the nearest integer. The rounding operation will increase $h(S) + g(S)$ by no more than a multiplicative factor.
>
> > Line 294: I am a little confused when the authors mentioned that (10) is less than or equal to (12). It seems that (12) should be a lower bound on the complexity (up to a constant) and the achieved time complexity (10) should not be smaller than it?
>
> (10) should be greater or equal to (12), and this is indeed the case. Note that we also have
> $$\frac{1}{\sqrt{m}} t^*(m) = \min_{j\in[n]} \left(\left(\sum_{i=1}^j \frac{1}{\tau_i}\right)^{-1} \left(\sqrt{m} + \frac{1}{\sqrt{m}} j\right)\right) \leq \min_{j\in[n]} \left(\left(\sum_{i=1}^j \frac{1}{\tau_i}\right)^{-1} \left(\sqrt{m} + j\right)\right) = t^*\left(\sqrt{m}\right).$$
> Together with the inequality in line 294, this demonstrates that Freya PAGE achieves the lower bound **up to a constant factor**. In other words, it holds that
> $$c_1 \times (12) \leq (10) \leq c_2 \times (12),$$
> where $c_1 \leq c_2$ are some universal constants.
>
> ---
>
> We trust that our responses have satisfactorily addressed the reviewer's questions. Should any additional clarifications be required, we are happy to provide them.

---

### Official Review · Reviewer_6ZMd · 2024-07-11

**Soundness:** 3
**Presentation:** 2
**Contribution:** 3
**Rating:** 5
**Confidence:** 3

**Summary:**

The paper introduces Freya PAGE, a novel parallel optimization method designed for distributed systems where computational resources vary in capability and speed. This method addresses the challenge of stragglers in asynchronous environments by adopting a strategy that adaptively ignores slower computations, thereby significantly improving the time complexity over previous methods like Asynchronous SGD and PAGE. The paper claims that Freya PAGE achieves optimal time complexity under weaker assumptions and also presents a theoretical analysis that establishes a lower bound for the time complexity in such settings. The approach is particularly effective in large-scale scenarios where the number of data samples is greater than the square root of the number of workers, thus demonstrating the algorithm's suitability for practical large-scale machine learning tasks.

**Strengths:**

1. The problem tackled is relevant to current large-scale machine learning applications, making the findings applicable and valuable to both academics and practitioners working with distributed computing environments.
2. Freya PAGE introduces a novel approach to handling heterogeneous and asynchronous computational environments in distributed systems. Its ability to adaptively ignore slower computations is a significant advancement over traditional methods.
3. The paper demonstrates that Freya PAGE achieves an optimal time complexity, which is a significant improvement compared to existing methods like asynchronous SGD.

**Weaknesses:**

1. Determining the optimal parameters $S$ and $p$ as outlined in Theorem 7 requires knowledge of unknown $\tau_i$ and solving an optimization problem, which can be challenging or even impractical.
2. The paper lacks sufficient experimental results on more complex machine learning tasks, including comparision with cutting-edge distributed learning algorithms.
3. The algorithm does not address the non-iid data senarios, workers may access data from varying distributions.

**Questions:**

How do the workers operate asynchronously? And why do the authors assert that Freya PAGE combines synchrony and asynchrony?

**Limitations:**

The authors have partially addressed the limitations.

---

> ### Author Rebuttal · Authors · 2024-08-04
>
> Thank you for your review and appreciating the strengths of our work. We would like to address your concerns and provide additional explanations.
>
> __Weaknesses__
>
> > Determining the optimal parameters $S$ and $p$ as outlined in Theorem 7 requires knowledge of unknown $\tau_i$ and solving an optimization problem, which can be challenging or even impractical.
>
> It is true that in general finding the optimal parameters requires access to $\{\tau_i\}$ (Theorem 6). However, in the large-scale regime (which is the main focus of the paper), the optimal $S$ and $p$ can in fact be determined very easily! The result in Theorem 7 states that the optimal parameters are $S^* = \lceil\sqrt{m}\rceil$ and $p^* = 1/\sqrt{m}$, where $m$ is the number of data points. Hence, no unknown parameters are involved. Importantly, the assumption that $\sqrt{m} \geq n$ in Theorem 7 is relatively weak, since the number of data points is typically much larger than the number of workers. Consequently, in practical scenarios, the optimal parameters can be found easily.
>
> > The paper lacks sufficient experimental results on more complex machine learning tasks, including comparison with cutting-edge distributed learning algorithms.
>
> We thank the reviewer for the suggestion. Given the theoretical nature of our work, our primary focus was not on extensive experimental validation. The empirical results included were intended to demonstrate that our theoretical findings align closely with practical observations, highlighting the robustness of our theoretical framework. Nonetheless, we are open to incorporating additional experimental results in the revised version of the paper.
>
> > The algorithm does not address the non-iid data scenarios, workers may access data from varying distributions.
>
> This is indeed an important scenario where appropriate asynchronous algorithms should be developed. However, establishing the time complexity even for the iid (homogeneous) data scenario was an open mathematical problem, which our paper resolves. Even in this scenario, we encountered non-trivial difficulties. Addressing these difficulties takes significant space and is enough to warrant a full research paper. We are currently considering the heterogeneous scenario, and we believe that this topic merits separate, dedicated work.
>
> __Questions__
>
> > How do the workers operate asynchronously? And why do the authors assert that Freya PAGE combines synchrony and asynchrony?
>
> Unlike fully asynchronous methods, Freya PAGE does not update the model based on gradients evaluated at stale parameters — this is the synchronous aspect of the update process, presented in Algorithm 1. However, at each iteration, Freya PAGE invokes the asynchronous subroutines ComputeGradient or ComputeBatchDifference. These subroutines handle the heterogeneity of processing times by waiting for the fastest worker to complete their computations and immediately assigning them a new job until a minibatch is computed. As a result, all workers are kept busy, and whenever they finish their computations, they asynchronously transmit the updates to the server.
>
> ---
>
> We believe that we have addressed all the issues raised by the reviewer and would appreciate if the reviewer could reconsidered the score. Please, let us know if any further clarifications are needed.

---

> > ### Comment · Reviewer_6ZMd · 2024-08-12
> >
> > Thank the authors for addressing my initial concerns.
> >
> > I notice that Freya SGD is quite similar to Rennala SGD [1] mentioned in the manuscript. While Freya SGD is designed for finite-sum optimization problems and Rennala SGD addresses stochastic programming, their adaptation from one context to another appears to be straightforward.
> >
> > Could the authors detail the distinct algorithmic and technical challenges that Freya SGD specifically addresses? This comparison would help delineate the unique contributions inherent in Freya SGD, particularly in how it navigates the complexities of finite-sum optimization compared to the broader scope of stochastic programming tackled by Rennala SGD.
> >
> > [1] Tyurin and Richtárik, “Optimal Time Complexities of Parallel Stochastic Optimization Methods under a Fixed Computation Model”, in NeurIPS 2023.

---

> ### Author Response · Authors · 2024-08-12
> **Official Comment by Authors**
>
> Thank you for the question.
>
> Indeed, in Line 304, we say that Freya SGD is closely related to Rennala SGD [1]. We consider Freya SGD to get the time complexity of the finite-sum optimization with a *non-variance reduced method.* Why is it important? First, such a method can be crucial and has a better convergence rate in the interpolation regime [2,3]. Second, while the dependence of Freya SGD on $\varepsilon$ is worse compared to Freya PAGE, the former method does not *explicitly* depend on $m$ (compare (10) vs the last formula in Theorem 25), which can be beneficial in some regimes. In total, Freya SGD provides a new time complexity that was not considered in the literature, which we believe is important for future work and for a general understanding of asynchronous methods.
>
> Let us explain the main differences between Freya SGD and Rennala SGD. The time complexity of Freya SGD is equal to
> $$\frac{\delta^0 L_-}{\varepsilon} \min_{j \in [n]} \left(\left(\sum_{i=1}^j \frac{1}{\tau_i}\right)^{-1} \left(\frac{L_{\max}}{\varepsilon} \left(\delta^0 + \Delta^*\right) + j \right)\right).$$
> Unlike Rennala SGD, this complexity does not depend on the noise $\sigma^2$ of stochastic gradients, which is a major advantage of Freya SGD. Another important difference between Rennala SGD and Freya SGD is the choice of the number of stochastic gradients that methods calculate in every iteration. In [1], the authors choose $S \approx \frac{\sigma^2}{\varepsilon},$ while we prove that the optimal choice of $S$ that minimizes the upper bound in Theorem 25 is $\approx \frac{L_{\max}}{\varepsilon} \left(\delta^0 + \Delta^*\right).$ Moreover, the convergence rate of Rennala SGD relies on the theory from [4], while Freya SGD uses the analysis with a different proof technique from [5].
>
> Beyond the main contributions related to Freya PAGE and the lower bound, we think that the analysis of Freya SGD is the cherry on top, offering deeper insights into time complexities in the finite-sum setting with heterogeneous asynchronous computations.
>
> We hope we have answered the question. If you need more information, we'd be happy to provide further details.
>
> [1] Tyurin and Richtárik, “Optimal Time Complexities of Parallel Stochastic Optimization Methods under a Fixed Computation Model”, in NeurIPS 2023.
>
> [2] M. Schmidt and N. L. Roux. Fast convergence of stochastic gradient descent under a strong growth condition. arXiv preprint arXiv:1308.6370, 2013.
>
> [3] S. Ma, R. Bassily, and M. Belkin. The power of interpolation: Understanding the effectiveness of SGD in modern overparametrized learning. In International Conference on Machine Learning, pages 3325–3334. PMLR, 2018.
>
> [4] Ghadimi, S. and Lan, G. (2013). Stochastic first-and zeroth-order methods for nonconvex stochastic programming. SIAM Journal on Optimization, 23(4):2341–2368.
>
> [5] A. Khaled and P. Richtárik. Better theory for SGD in the nonconvex world. Transactions on Machine Learning Research, 2022.

---

> > ### Author Response · Authors · 2024-08-13
> > **Official Comment by Authors**
> >
> > For some reason, OpenReview did not send an email notification for our previous post above. This comment is sent to try to activate another email notification.

---

> ### Author Response · Authors · 2024-08-13
> **Optimality**
>
> We wish to re-iterate that despite decades of research on SGD (SGD was studied at least since 1951; Robbins and Monroe) and on asynchronous methods (1986; Tsitsiklis et al), FreyaPAGE is the first provably optimal asynchronous SGD method for minimizing finite sums in the smooth nonconvex data-homogeneous regime.
>
> We are biased of course, but we believe such a result should command a substantially higher score than 5, despite the three shortcomings mentioned by the reviewer. We addressed the first weakness in our rebuttal - please check our response. We view the second weakness as minor since our work is of a theoretical nature. We believe theoretical works should be judged on the basis of the strength of the theory, just like empirical works should be judged based on the experimental results. Nevertheless, we will include several additional well-designed experiments in the camera-ready version of the paper. Lastly, we acknowledge the third weakness. Having said that, our work is a first step to considering more challenging scenarios in the future, by us or others in the community.
>
> Of course, we understand the reviewer may have a different view. Nevertheless, we thought it might be useful to say how we feel about the theoretical importance of our result. Thanks again for your review and for considering our rebuttal and responses!
>
> Authors

---

### Official Review · Reviewer_QBDK · 2024-07-12

**Soundness:** 3
**Presentation:** 4
**Contribution:** 3
**Rating:** 8
**Confidence:** 3

**Summary:**

The paper introduces Freya PAGE, a new parallel method for large-scale nonconvex finite-sum optimization in heterogeneous and asynchronous computational environments. Freya PAGE, specifically addresses the variability in processing times across different workers due to hardware and network differences. By being robust to "stragglers" and adaptively ignoring slow computations, Freya PAGE  improves time complexity guarantees compared to existing methods. Additionally, the paper proves a lower bound for smooth nonconvex finite-sum problems in asynchronous setups.

**Strengths:**

Freya PAGE is a novel parallel optimization method specifically designed to handle heterogeneous and asynchronous computations. This appears to be a significant advancement and addresses real-world challenges in distributed systems. By addressing the variability in processing times across different workers due to hardware and network differences, the method is highly relevant to practical distributed systems used in large-scale machine learning tasks. This method shows improved time complexity guarantees compared to existing methods. The paper also provides a strong theoretical framework in general. Specifically, proving lower bound for smooth nonconvex finite-sum problems in asynchronous setups shows the optimality of Freya PAGE in large-scale regimes. The stochastic gradient collection strategies ComputeGradient and ComputeBatchDifference introduced in this paper is innovative. Overall, this is a very well written paper and easy to follow and understand.

**Weaknesses:**

The empirical results of this paper are primarily focused on synthetic quadratic optimization tasks and logistic regression problems. More diverse and extensive experimentation on a wider range of real-world datasets and applications would strengthen the practical validation.

The algorithm's performance might vary significantly for different configurations of worker speeds and hardware setups. The variance in performance could be an issue in highly dynamic environments.

**Questions:**

If you have tested Freya PAGE in real-world scenarios, can you share the results and insights?
How does Freya PAGE specifically determine which computations to ignore or adaptively manage in the presence of stragglers? Can this mechanism be fine-tuned for different distributed system configurations?

**Limitations:**

The performance of Freya PAGE might vary significantly with different configurations of worker speeds and hardware setups. This variance could be problematic in environments with highly variable computational resources.
The impact of communication overhead in distributed systems with significant network latency, is not extensively discussed.

---

> ### Author Rebuttal · Authors · 2024-08-04
>
> Thank you for your feedback and for recognizing the strengths of our work. We would like to address each of your comments and provide clarifications.
>
> __Weaknesses__
>
> > The empirical results of this paper are primarily focused on synthetic quadratic optimization tasks and logistic regression problems. More diverse and extensive experimentation on a wider range of real-world datasets and applications would strengthen the practical validation.
>
> We thank the reviewer for the suggestion. Given the theoretical nature of our work, our primary focus was not on extensive experimental validation. The empirical results included were meant to demonstrate that our theoretical findings align closely with practical observations, underscoring the robustness of our theoretical framework. Nevertheless, we are open to incorporating additional experimental results.
>
> > The algorithm's performance might vary significantly for different configurations of worker speeds and hardware setups. The variance in performance could be an issue in highly dynamic environments.
>
> It is certainly true that the more challenging the problem and the less favorable the environment, the worse the performance of any algorithm. However, it is important to note that our method matches the lower bound established in Theorem 10, demonstrating that Freya PAGE is optimal for the problem we consider, and suggesting that it is infeasible to develop a method with a better performance.
>
> __Questions__
>
> > How does Freya PAGE specifically determine which computations to ignore or adaptively manage in the presence of stragglers? Can this mechanism be fine-tuned for different distributed system configurations?
>
> The greatest strength of the algorithm is its ability to adapt to the behavior of clients automatically, without requiring manual fine-tuning of this mechanism. Specifically, consider Algorithms 2 and 3 (the asynchronous subroutines of Freya-PAGE). In step 6, the algorithm waits for any worker to send any update. As soon as an update is received, the gradient estimator is updated, and the worker is immediately assigned a new job. This mechanism operates automatically and independently of the order in which messages arrive - whichever worker completes their task first has its update used and is given new data to process. If some workers are too slow to participate in the training, their computations can potentially never be completed, and hence they may effectively be ignored. However, the algorithm itself does not ignore any messages it receives. This is the reason why Freya PAGE is capable of fully utilizing the available computing resources, and why it achieves its superior performance.
>
> __Limitations__
>
> > The performance of Freya PAGE might vary significantly with different configurations of worker speeds and hardware setups. This variance could be problematic in environments with highly variable computational resources.
>
> Please refer to our responses to the weaknesses above.
>
> > The impact of communication overhead in distributed systems with significant network latency, is not extensively discussed.
>
> In this paper, we do not directly focus on the communication overhead. However, our method is communication-friendly. As noted just after Algorithms 1 and 2, _the workers can aggregate $\nabla f_p$ locally, and the algorithm can call AllReduce once to collect all calculated gradients_. It means that Freya PAGE requires each worker participating in the training to send one vector per iteration/round.
>
> It is possible to improve the communication overhead further using, for instance, compression techniques. It is a research direction that warrants a separate study. We therefore leave this topic for future work.
>
> ---
>
> We are grateful for your feedback and believe that we have addressed the questions. If further information or clarifications are required, please do not hesitate to ask.

---

> > ### Comment · Reviewer_QBDK · 2024-08-10
> > **On the rebuttal**
> >
> > I thank the authors for their responses. I have read the responses carefully and I have no further questions. I am inclined to retain my original scores.

---

### Official Review · Reviewer_MLsi · 2024-07-13

**Soundness:** 3
**Presentation:** 4
**Contribution:** 3
**Rating:** 6
**Confidence:** 3

**Summary:**

This paper considers the problem of minimizing a finite sum of non-convex objective terms with multiple computational units that calculate gradient oracles. It introduces two efficient computational subroutines for the main steps of the PAGE algorithm, and provides novel bounds on the real time of computation assuming a heterogeneous computational scenario. For this, the paper gives bounds on the execution time of these subroutines and combines them with the previously known iteration complexity of PAGE. Next, the paper considers multiple variations: it optimizes the hyperparameters of the PAGE algorithm in different regimes, and also extends the results to a dynamic scenario. Finally, it provides a lower bound on the overall execution time and in this way establishes that their strategy is optimal in large-scale regimes.

**Strengths:**

The paper is well written and has an excellent presentation of the subject. It provides an extensive theoretical study of a novel gradient management strategy in the PAGE algorithms, and achieves tight bounds on the convergence rate.

**Weaknesses:**

I do not have any major concern about this work. The only limitation for me is that from a practical standpoint, this algorithm might not be optimal as it may introduce a significantly higher communication overhead. This is because each gradient is now required to be sent in an individual message.

**Questions:**

Compared to the results in the appendix, the stated theorems are generally simplified for better comprehension, but some confusion is also introduced. In particular, I am not sure which results are deterministic and which results are in terms of statistical expectation or high-probability. Could you clarify?

In the statement of Theorem 4, you assume two bounds on m, where one of them seem to be stronger than the other. Why do you keep both?

**Limitations:**

No limitations or potential negative societal impact.

---

> ### Author Rebuttal · Authors · 2024-08-04
>
> Thank you for your review and appreciating the strengths of our work. We would like to address your comments and provide additional explanations.
>
> __Weaknesses__
>
> > The only limitation for me is that from a practical standpoint, this algorithm might not be optimal as it may introduce a significantly higher communication overhead. This is because each gradient is now required to be sent in an individual message.
>
> Notice that it is possible to avoid sending an individual message for each computed gradient. As noted under Algorithms 1 and 2, _the workers can aggregate $\nabla f_p$ locally, and the algorithm can call AllReduce once to collect all calculated gradients_. While, indeed, the current listings of Algorithms 1 and 2 require the workers to send individual gradients, this strategy can be slightly improved. We can ask the workers to aggregate the gradients locally and send these aggregated vectors at the end of Algorithms 1 and 2. This simple modification can significantly reduce the communication overhead.
>
> It is possible to improve the communication complexity of the algorithm even further, e.g., through compression techniques. It is a research direction that deserves a separate study. We therefore leave this topic for future work.
>
> __Questions__
>
> > Compared to the results in the appendix, the stated theorems are generally simplified for better comprehension, but some confusion is also introduced. In particular, I am not sure which results are deterministic and which results are in terms of statistical expectation or high-probability.
>
> The algorithms presented in Algorithms 1, 2, and 3 are all stochastic in nature, which means that the complexity results are not deterministic. The iteration complexities (Theorem 4) are derived in terms of the expected number of iterations needed to find an $\varepsilon$-stationary point. Similarly, Theorems 1 and 2 describe the expected time required to compute the relevant quantities (hence the term 'expected' in Theorem 1; this word is indeed omitted in some other results - we will correct this oversight in the revised version of the paper). The time complexities outlined in Theorems 5, 6, 7, and 8 are based on expected iteration complexities, and thus are also non-deterministic. Thank you for pointing this out, we will make the adjustments in the paper.
>
> > In the statement of Theorem 4, you assume two bounds on m, where one of them seem to be stronger than the other. Why do you keep both?
>
> We believe that the reviewer is referring to Theorem 7 (as there are no bounds on $m$ in Theorem 4; please correct us if a different result was meant). The result in Theorem 7 states that $S^* = \lceil\sqrt{m}\rceil$ and $p^* = 1/\sqrt{m}$ are the optimal parameters when $\sqrt{m}\geq n$. The inequality $m \geq n\log n$ comes from Assumption 4, and does not need to be included. Thank you for pointing this out.
>
> ---
>
> We are grateful for the valuable feedback and trust that our responses have satisfactorily addressed the reviewer's concerns. Please let us know if more details or clarifications are needed.

---

> > ### Comment · Reviewer_MLsi · 2024-08-10
> >
> > Many thanks for considering my comments. I maintain my score.

---

### Decision · Program_Chairs · 2024-09-25

**Decision:**

Accept (poster)

**Comment:**

This paper proposes a distributed stochastic optimization algorithm called Freya Page that is suitable for heterogeneous asynchronous computations. It is shown that for general nonconvex problems, the gradient converges to zero in expectation at a rate that approaches the lowerbound in the large scale regime. The reviewers all appreciate the theoretical contributions of this work. Some minor concerns include limited experimental results and potential difficulty in parameter tuning in practice.